# Evaluation of an emergent feature of sub-shelf melt oscillations from an idealised coupled ice-sheet/ocean model using FISOC(v1.1)-ROMSIceShelf(v1.0)-Elmer/Ice(v9.0)

Chen Zhao [1], Rupert Gladstone [2], Benjamin Keith Galton-Fenzi [3,1,4], David Gwyther [5,6], and Tore Hattermann [7,8]

[1]Australian Antarctic Program Partnership, Institute for Marine and Antarctic Studies, University of Tasmania, Hobart, Australia
[2]Arctic Centre, University of Lapland, Rovaniemi, Finland
[3]Australian Antarctic Division, Kingston, Australia
[4]The Australian Centre for Excellence in Antarctic Science, University of Tasmania, Hobart, Australia
[5]Centre for Applications in Natural Resource Mathematics, School of Mathematics and Physics, University of Queensland, Australia
[6]Coastal and Regional Oceanography Lab, School of Biological, Earth and Environmental Sciences, UNSW Sydney, Australia
[7]Norwegian Polar Institute, Tromsø Norway
[8]Energy and Climate Group, Department of Physics and Technology, The Arctic University - University of Tromsø, Norway

**Correspondence:** Chen Zhao (Chen.Zhao@utas.edu.au)

**Abstract.** Changes in ocean-driven basal melting have a key influence on the stability of ice shelves, the mass loss from the ice sheet, ocean circulation and global sea level rise. Coupled ice sheet - ocean models have a critical role in understanding future ice sheet evolution and examining the processes governing ice sheet response to basal melting. However, as a new approach, coupled ice-sheet/ocean systems come with new challenges, and the impacts of solutions implemented to date have not been investigated. An emergent feature in several contributing coupled models to the 1st Marine Ice Sheet–Ocean Model Intercomparison Project (MISOMIP1) was a time-varying oscillation in basal melt rates. Here we use a recently developed coupling framework, FISOC (v1.1), to connect the modified ocean model ROMSIceShelf (v1.0) and ice-sheet model Elmer/Ice (v9.0), to investigate the origin and implications of the feature and more generally the impact of coupled modelling strategies on the simulated basal melt in an idealised ice shelf cavity, based on the MISOMIP setup. We found the spatial-averaged basal melt rates (3.56 m yr$^{-1}$) oscillated with an amplitude $\sim 0.7$ m yr$^{-1}$ and approximate period of $\sim 6$ years between year 30 and 100, depending on the experimental design. The melt oscillations emerged in the coupled system and the stand-alone ocean model using a prescribed change of cavity geometry. We found that the oscillation feature is closely related to the discretised ungrounding of the ice sheet, exposing new ocean, and is likely strengthened by a combination of positive buoyancy-melt feedback and/or melt-geometry feedback near the grounding line, and the frequent coupling of ice geometry and ocean evolution. Sensitivity tests demonstrate that the oscillation feature is always present, regardless of the choice of coupling interval, vertical resolution in the ocean model, tracer properties of cells ungrounded by the retreating ice sheet, or the dependency of friction velocities to the vertical resolution. However, the amplitude, phase and sub-cycle variability of the oscillation varied significantly across the different configurations. We were unable to ultimately determine whether the feature

arises purely due to numerical issues (related to discretization), or a compounding of multiple physical processes amplifying a numerical artifact. We suggest a pathway and choices of physical parameters to help other efforts understand the coupled ice-sheet/ocean system using numerical models.

## 1 Introduction

Antarctica is surrounded by floating ice shelves that link the grounded continental ice sheet with the oceans. Ice shelf changes play a significant role in the sea level rise by altering the discharge of grounded ice into the ocean due to the buttressing effect of the ice shelf on the grounded inland ice. A significant proportion of the ice mass loss from ice shelves is through basal melting driven by ocean-warming (Rignot et al., 2013; Liu et al., 2015), which is mainly controlled by the oceanic thermodynamic and circulatory processes in the ice shelf cavity. Additionally, glacial meltwater from the ice shelves affects sea ice formation, ocean circulation, water mass transformations as well as the creation of Antarctic Bottom Water, which influences the global ocean and climate (e.g., Potter and Paren (1985), Jacobs and Giulivi (2010), Beckmann and Goosse (2003),Hellmer (2004),Hellmer et al. (2017)). Understanding the ice-ocean interaction beneath the ice shelf is essential for interpreting the past and recent changes on the ocean and ice sheets and for predicting the future of the Antarctic Ice Sheet and its impact on global sea level and the climate system.

Direct and fine-scale oceanographic observations in the cavities beneath Antarctic ice shelves are logistically challenging and numerical models that include the ice/ocean interaction has been playing an invaluable role in investigating the processes governing basal melting (Dinniman et al., 2016). Over the last few years, several three-dimensional ocean models have been developed to be coupled to ice sheet models for presenting a moving ocean-ice boundary, for example see ROMS: Galton-Fenzi et al. (2012); FVCOM: Zhou and Hattermann (2020); NEMO: Smith et al. (2021), Favier et al. (2019), Mathiot et al. (2017); MITgcm: Jordan et al. (2018), Goldberg et al. (2018), De Rydt and Gudmundsson (2016); POP2x: Asay-Davis et al. (2016). These models provide efficient ways to simulate the small or large-scale spatial and temporal evolution of the basal melting beneath the Antarctic ice shelves in idealised configurations.

The development of coupled ice sheet/ocean model motivated various model intercomparison projects to evaluate the assets of models and their physics through idealistic configurations, including the stand-alone components (MISMIP+ and ISOMIP+) and coupled models (MISOMIP1) (Asay-Davis et al., 2016). The first phase of the Marine Ice Sheet–Ocean Model Intercomparison Project (MISOMIP1) is a community effort aimed at evaluating coupled ice sheet-ocean systems with idealised topography and forcing to explore and better understand ice-ocean interactions for key regions of the West Antarctic Ice Sheet (Asay-Davis et al., 2016). Initial results indicate the simulated ocean-driven basal melting displays differences in magnitudes and patterns between models. A similar oscillation pattern in the simulated ocean-driven basal melting rates has been demonstrated by most of the coupled models in MISOMIP1 (Fig. 1; Asay-Davis, personal communication), which is not understood to be an emergent physical property or a numerical artefact. The coupled model NEMO-Elmer/Ice also generated the similar pattern while the parameterised simulations with a fixed ice draft produce melt rates without the same oscillations (Favier et al., 2019). There are many physical and numerical parameters affecting the ice-ocean interaction at the interface, which

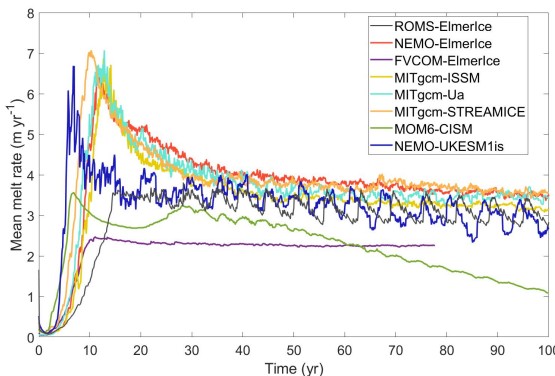

**Figure 1.** Simulated mean melt rates from different coupled models in MISOMIP1 project (pers. comm. Xylar Asay-Davis).

might cause such oscillation feature but have not been explored in previous studies. Research found that vertical discretisation and resolution of ocean model significantly affect the basal melt rate through differences in how the meltwater fluxes are distributed and how the thermal driving is calculated (Gwyther et al., 2020). Higher resolution of the ice-ocean boundary
layer region in terrain-following ocean models, e.g. ROMS, may simulate lower melt rates compared to Z-coordinate models (Gwyther et al., 2020). The coupled system can also be sensitive to the choice of timestep, e.g. the split time stepping scheme used in ROMS (Shchepetkin and McWilliams, 2005), the coupling interval, and the timestep in the ice model.

However, it is still unknown about how the parameterisations of the coupled system would affect the simulation of the thermodynamic exchange of heat and freshwater occurred in the ice-ocean boundary. Here we assess the sub-shelf melting
parameterisations in the coupled system and explore the possible origins of the oscillations within a coupled model framework.

In this study we use the Framework for Ice Sheet-Ocean Coupling (FISOC) (Gladstone et al., 2021) to couple the Elmer/Ice (Gagliardini et al., 2013) and a modified version of the Regional Ocean Modeling System (ROMSIceShelf, Galton-Fenzi et al. (2012)) to model ice shelf/ocean interactions for an idealised three-dimensional domain. Experiments in this paper follow the protocol of MISOMIP1 which is fully described in Asay-Davis et al. (2016). A series of experiments are motivated by the
direct coupling relationship illustrated in the initial MISOMIP1 results (e.g. Favier et al., 2019), the impact of coupling interval on simulating the basal melt rates, whether the vertical resolution of ocean model have significant effect on basal melting in the coupled system as indicated in Gwyther et al. (2020), the dependency of basal melt parameterisation on the friction velocity, and how wet/dry cells are handled in response to grounding line (GL) movement.

## 2 Model Setup and Experiment Design

### 2.1 Initial geometry

The domain used here is the MISOMIP1 setup (Fig. 2 Asay-Davis et al., 2016). The domain is 800 km long in north-south direction (with x positive northward) and 80 km wide in the east-west direction (with y positive westward) with 1 km horizontal

resolution in the ice component and 2 km in the ocean component. The initial ice sheet is built within the framework of MISMIP+, after spinning-up for 215,000 yrs from a uniform ice thickness of 100 m without basal melting, and is thus in equilibrium state (Cornford et al., 2020). The ice sheet rests on a retrograde bed sloping upward towards the ocean with the initial floating ice front at 640 km.

## 2.2   Experiment design

Each coupled model experiment in this study was run for 100 years, following Experiment IceOcean1r of MISOMIP1 Asay-Davis et al. (2016). Like in IceOcean1r, experiments in this study does not include a dynamic calving, in which ice thickness is allowed to be zero without calving. Various configuration in each experiment can be seen in Table 1 and corresponding sections in Sec. 3.

We build our coupled model following the ISOMIP+ projects for stand-alone ocean models with ice-shelf cavities and the MISMIP+ projects for ice sheet models. Result of ISOMIP+ Ocean3 from Asay-Davis et al. (2016) using the same ocean model will be used as a comparison to the control experiment in this study (CTRL in Table 1).

The ocean model in the coupled system is initialised with a steady-state ice geometry from the ice sheet model and a "COLD" initial condition following Asay-Davis et al. (2016). No external forcing is applied at the surface of the open ocean, which means there is no atmospheric or sea-ice fluxes. A "WARM" forcing, as the only forcing, is applied within a 10 km restoring region near the ocean's northern boundary (yellow area in Fig. 2a), which is consistent with the warm ice shelf cavities in Amundsen and Bellingshausen Seas. The warm water is expected to reach the ice-shelf cavity within the first two decades and induce strong basal melting and subsequent rapid GL retreat.

In Ocean3, the stand-alone ocean model uses the same steady-state ice topography with the initial state of the coupled system, and is run for 100 years with an annually prescribed evolving ice geometry. The ocean is initialized with the WARM profiles, forced with the WARM profile in the same restoring region with CTRL and strong melting is expected to begin immediately as the sub-shelf circulation spins up. More details about MISMIP+ and ISOMIP+ can be seen in Asay-Davis et al. (2016).

## 2.3   The ice-sheet model, Elmer/Ice

The ice component uses Elmer/Ice, a three-dimensional (3-D), finite-element, dynamic ice model (Gagliardini et al., 2013). Elmer/Ice is able to directly solver the Stokes equations, and also offers multiple options to approximate the Stokes equations. Here, we use the SSA* solution, a variant of the L1L2 solution of Schoof and Hindmarsh (2010), to solve the shallow shelf approximation of the Stokes equations. The SSA* approximation includes longitudinal and lateral stresses and an assumption of a simplified vertical shear profile that represents fast-flowing ice streams and ice shelves. A constant ice temperature is assumed in this study and the thermal conductivity of ice is equal to zero, which means there is no heat flux into the ice at the boundaries. Ice that is lost due to calving disappears immediately and does not produce a freshwater flux into the ocean. We applied a non-linear Weertman-type sliding relationship (Eq. (21) in Gagliardini et al. (2013)) with a sliding parameter equal to 0.01 and an exponent equal to 1/3. The other parameters used in the ice model follow Table 1 in Asay-Davis et al. (2016).

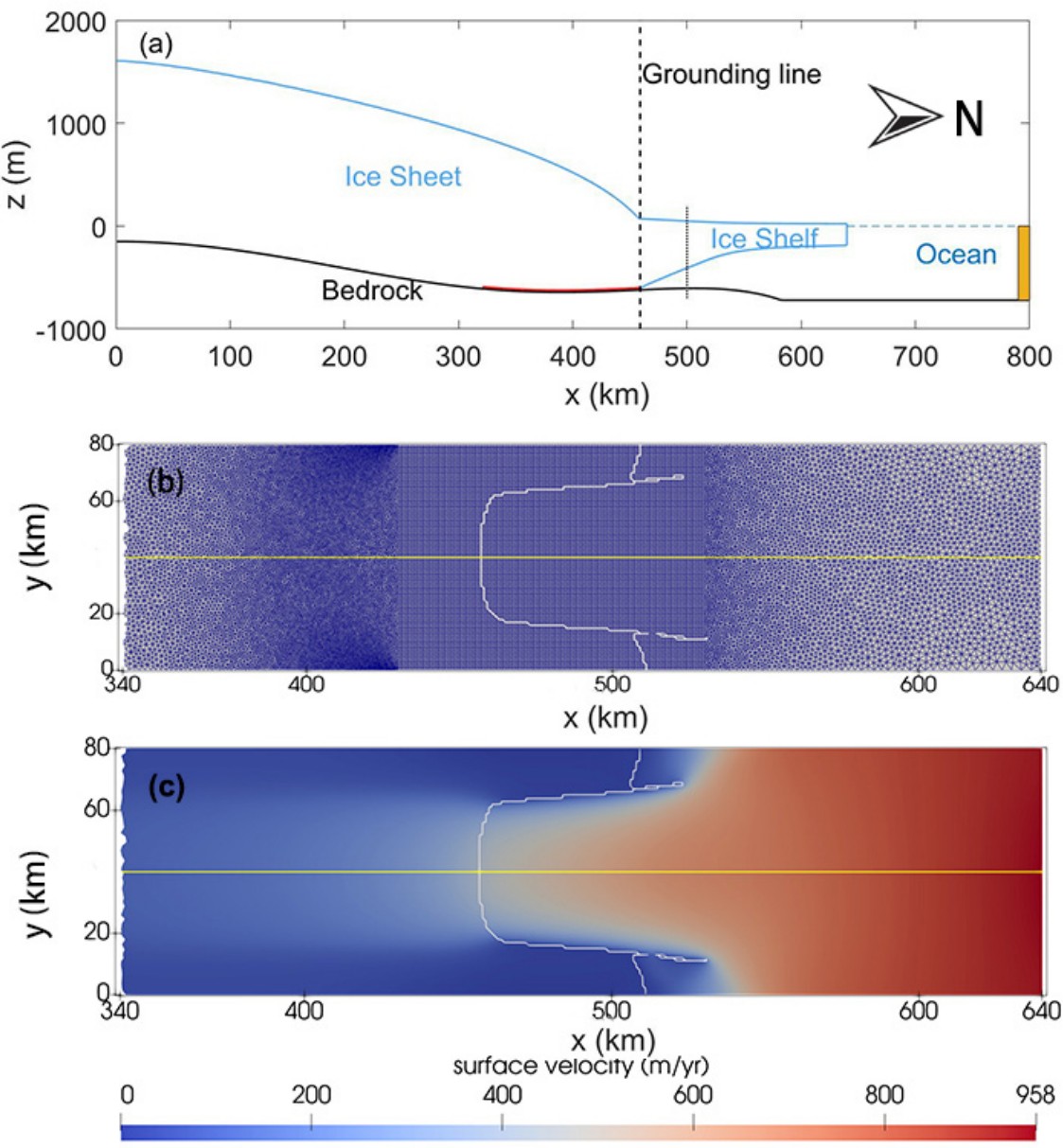

**Figure 2.** Schematic of the initial geometry. (a) The geometry profile in the xz plane along the central flow line (yellow line in panel (b)). The red line is the zone where the wet-dry cells are dry (see Sec. 2.5 for details). The yellow filled area is the restoring region. (b) The mesh of ice model in the xy plane with the grounding line shown as the white line.(c) Magnitude of surface velocity in the xy plane.

## 2.4 The ocean model, ROMSIceShelf

The ocean component uses a modified version of the Regional Oceanic Modeling System (ROMSIceShelf), a 3D terrain-following (s-coordinate) numerical ocean model that has been developed for ice shelf cavity modelling studies (e.g. Galton-Fenzi et al., 2012; Galton-Fenzi, 2019). ROMS uses a stretched terrain-following vertical coordinate that we apply with higher resolution near the ocean surface and near to the sea bed, to better resolve surface and bottom boundary layers (e.g. Dinniman et al., 2007; Galton-Fenzi, 2019; Gwyther et al., 2020).

The vertical grid for the control experiment uses 21 vertical layers with the top-layer cell thickness equal to 0.5 m adjacent to the GL, about 2 m at mid-ice shelf and about 5.5 m near the ice front. The bottom-layer cell thickness is equal to 1.2 m adjacent to the GL, about 13 m at mid-ice shelf and about 32 m near the ice front. The other parameters used in the ocean model follow Table 4 in Asay-Davis et al. (2016).

ROMS uses a split time stepping scheme, where the two-dimensional, vertically integrated momentum equations are solved using a short barotropic timestep to resolve barotropic waves, and the three-dimensional momentum equations are solved using a longer baroclinic timestep (Ezer et al., 2002). Oceanic model solutions can be sensitive to the choice of timestep (e.g. Shchepetkin and McWilliams, 2005) so we first performed preliminary experiments to decide on the appropriate timestep to be used in the ocean model by the control experiment when testing the coupled model. The timestep used in the ice component equal the coupling interval here, which will be explored in Sec. 3.2.

Results from the preliminary suite of ocean model timestep experiments demonstrate that large-scale response of the simulated mean melt rate is relatively insensitive to the choice of barotropic and baroclinic timestep (Fig. A1). Additionally, it indicates that the barotropic timestep sizes has a dominant effect on simulating the basal melting (Fig. A1). Experiments with same barotropic timestep (e.g., batrotropic DT = 1.67 s for experiments DT50N30, DT100N60, DT200N120 in Table A1) produce a similar response in the spatial-averaged melt rates while tests with same baroclinic timestep size but different barotropic timestep sizes (e.g., DT200N30 = 6.67 s, DT200N60 = 3.33 s, DT200N120 = 1.67 s A1) show subtle variations in the spatial-averaged basal melting. Experiments with smaller barotropic timestep (DT50N30, DT100N60, DT200N120) produce a relatively smoother and lower spatial-averaged melt rate, with less noise than other experiments. Here we have also tested the influence on the choice of ratios between the timesteps with negligible result (see Appendix A). Note that the spin-up period required for the models to approach equilibrium was about the same for all experiments.

Based on these preliminary timestep experiments we used DT100N60 as the control (CTRL in Table 1), given this choice also provides computational efficiency (about 576 cpu-hours for 1 year coupling time run on 96 processors for NCI's Gadi supercomputer). For the unstable experiments in this study, a smaller baroclinic timestep (DT50N30) was used for resolving instabilities and producing a simulation that runs to completion.

The thermodynamic exchange of heat at the ice-ocean interface is parameterised with the standard 'three-equation parameterisation' (Hellmer and Olbers, 1989; Holland and Jenkins, 1999; Asay-Davis et al., 2016). Here we discuss some of the specific components that are relevant to this study but refer the reader to Hellmer and Olbers (1989); Holland and Jenkins (1999); Asay-Davis et al. (2016) for more information.

The water speed in the three-equation parameterisation explicitly includes a constant tidal offset, $u'^2_w = u_w{}^2 + u_{tidal}{}^2$. $u_{tidal}$ is the root mean square velocity associated with tide. Here we set it as a constant value of 0.01 $m/s$. $\Gamma_T$ and $\Gamma_S$ are the constant non-dimensional heat- and salt-transfer coefficients. Other parameters are defined in (Jenkins et al., 2010), with subscripts $_{i,b}$ $_{fw}$ and $_{sw}$ representing ice, ice-ocean interface, freshwater and seawater, respectively. The three-equation parameterisation is typically applied between the top model layer and the ice, and hence the temperature to drive melting and the depth over which heat and freshwater fluxes from melt are released will change as a function of the vertical resolution.

In addition to the thermodynamic exchange at the ice-ocean interface, the pressure at the ice-ocean interface as an ocean boundary condition in the ocean model is calculated using the ice draft and a constant reference ocean density (1028 $\mathrm{kg\,m^{-3}}$).

Previous studies found that higher resolution of the ice-ocean boundary layer region in terrain following ocean models, e.g. ROMS, may produce lower melt rates as compared with Z-coordinate models (Gwyther et al., 2020). Gwyther et al. (2020) found that ocean models with different patterns for distributing meltwater fluxes and sampling tracers for melting did not make much difference when the effective vertical resolution adjacent to the ice base is similar. In this study, the impact of vertical resolution on simulating basal melt in a coupled system is explored in Sec. 3.3.

Turbulence generated by velocity shear in the boundary layer is important for transferring heat and salt to the ice base (Holland and Jenkins, 1999). Here we adopt a simple parameterisation of the boundary layer by defining the surface shear stress as a quadratic function using the nearest cells' current speed, which is used to calculate the friction velocity, $u_*$. The choice of the nearest currents may introduce a resolution dependency due to the way of sampling of the representative velocity (Gwyther et al., 2020). In ROMS the nearest currents are sampled in the top ocean cell (Gwyther et al., 2020), while the friction velocity in the z-coordinate models is calculated with the mean velocity over a prescribed distance from the ice (Losch, 2008). Resolution dependency in the method for calculating $u_*$ is also explored in the coupled system in Sec. 3.4.

Changes in water column thickness due to ice shelf thinning would be maintained through increased horizontal convergence/divergence in the ocean circulation in response to mass/volume changes. ROMS effectively introduces a source/sink term imposed by adding or removing heat or salt at the ice/ocean boundary. For example, when the ice shelf melts, the model removes salt/heat rather than adding freshwater volume. The circulation change in this case is a result from density changes rather than volume changes. The approach using a source/sink term of heat/salt transfer imposes a choice upon the ocean model: either conserving the volume integrals of tracer values (temperature and salt) or preserving the absolute values, (e.g., heat or freshwater). Here we will explore the effect of both options on the ocean circulation in a coupled system in Sec. 3.5.

## 2.5 The coupling framework, FISOC

FISOC is an open source flexible coupling framework built based on the existing Earth System Modelling Framework (ESMF, (Hill et al., 2004); (2005)) and provides a modular approach to facilitate using combinations of different ice and ocean models for application to Antarctic ice sheet - ocean systems (Gladstone et al., 2021). Simulations in this study use bilinear regridding method provided by ESMF and 'Corrected rate' option for the cavity evolution (Gladstone et al., 2021). The coupling interval is set to be same with the timestep size in the ice component (15 days in the control experiment), while the time resolution of the ocean component is much finer (100 s). Semi-synchronous coupling is adopted here, in which the ice component has a

**Table 1.** Summary of the experiments used in this study. NEAREST means the tracer properties of the dry cells are set to be equal to the nearest wet cell. 21E means 21 vertical layers with equal layer thickness. CDT means the coupling interval. FMT means the flux mixing thickness representing the depth over which meltwater fluxes are mixed or distributed. FMT is set to 20 m in FMT20 while in other experiments equals the spatially averaged top layer thickness. PIT means the volume integral of tracer properties are preserved while PAT means the absolute tracer properties are preserved. TOP means the friction velocity is calculated based on the velocity of the top model cell. MEAN means the friction velocity is calculated from the mean velocity of the top three model cells. INDEP assumes that the tracer property transfer is independent from the friction velocity.

| Simulation | CDT (days) | Vertical layers | FMT (m) | Tracer properties | Conservation Strategy | Ustar |
|---|---|---|---|---|---|---|
| CTRL | 15 | 21 | 2.8 | NEAREST | PAT | TOP |
| CDT90 | 90 | 21 | 2.8 | NEAREST | PAT | TOP |
| CDT30 | 30 | 21 | 2.8 | NEAREST | PAT | TOP |
| CDT5 | 5 | 21 | 2.8 | NEAREST | PAT | TOP |
| CDT1 | 1 | 21 | 2.8 | NEAREST | PAT | TOP |
| CDT0.5 | 0.5 | 21 | 2.8 | NEAREST | PAT | TOP |
| WETDRY1 | 15 | 21 | 2.8 | T=-1.8°C S=33.8 PSU | PAT | TOP |
| WETDRY2 | 15 | 21 | 2.8 | T=-0.8°C S=34.2 PSU | PAT | TOP |
| WETDRY3 | 15 | 21 | 2.8 | T=-1.3°C S=34.0 PSU | PAT | TOP |
| PIT | 15 | 21 | 2.8 | NEAREST | PIT | TOP |
| N11 | 15 | 11 | 9.2 | NEAREST | PAT | TOP |
| N21E | 15 | 21E | 2.8 | NEAREST | PAT | TOP |
| FMT20 | 15 | 21 | 20 | NEAREST | PAT | TOP |
| UstarMean | 15 | 21 | 2.8 | NEAREST | PAT | MEAN |
| UstarIndep | 15 | 21 | 2.8 | NEAREST | PAT | INDEP |

larger time step than the ocean model. We set the coupling interval equal to the ice model timestep size (15 days in CTRL), while the baroclinic timestep in the ocean model is 100 s.

In ROMSIceShelf, the GL position movement is based on the evolving cavity geometry passed from Elmer/ice through FISOC. Here we use a "wet-dry" scheme (similar to the "thin film" approach in Goldberg et al. (2018)) to allow the GL movement. A thin passive water layer of 20 m is created between the grounded ice and bed (see the red line in Fig. 2). "Dry" cells represent the passive water column under grounded ice while the "wet" cells represent the active water column beneath floating ice or the open ocean. An activation criterion for an "dry" cells turning into "wet" is imposed to represent GL retreat.

If dynamic variations in ocean pressure are sufficient to overcome the ice pressure due to the positive height above buoyancy, the dry cell unground and become wet. More detailed information can be seen in Gladstone et al. (2021).

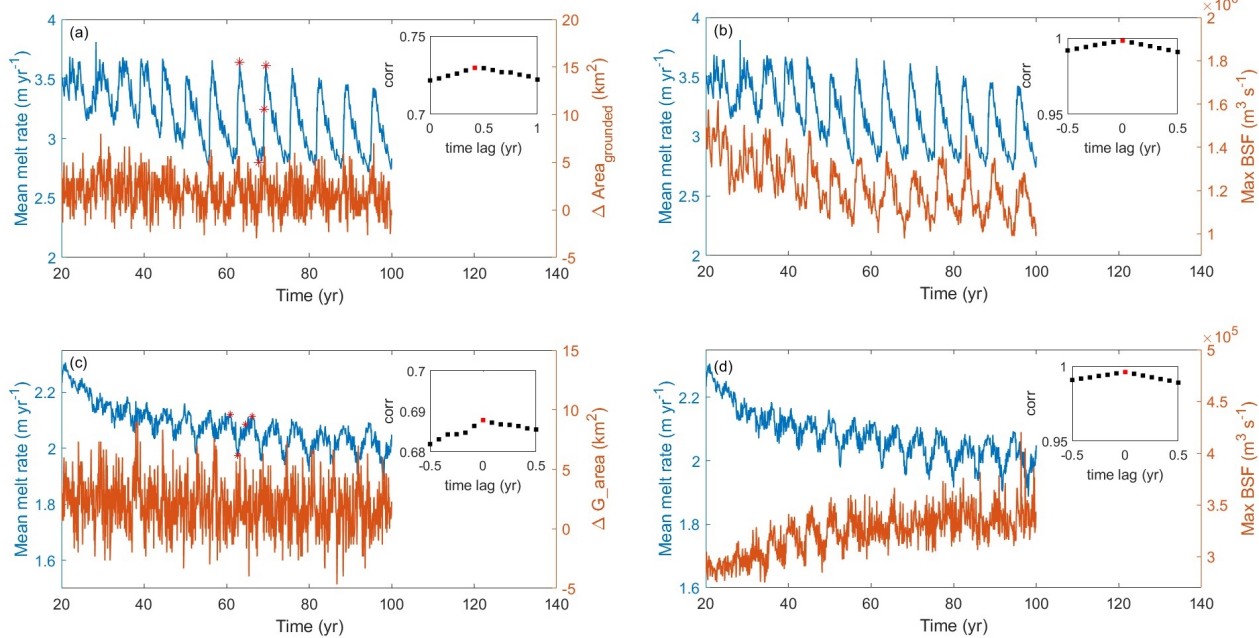

**Figure 3.** Mean melt rate and the moving mean changes in grounded area from (a) CTRL and (c) Ocean3, mean melt rate and maximum of the barotropic stream function from (b) CTRL and (d) Ocean3 starting from year 20. The correlation coefficients between the blue line and the orange line are shown in the inset of each panel with the red dots indicating the highest correlation coefficient. A normalized passband frequency 0.90 is applied to remove noise.

## 3 Results

### 3.1 Oscillations of the spatial-averaged basal melt rate

The blue line in Fig. 3a shows the simulated mean melt rates from CTRL starting from year 20 and the result with first 20 years
is shown in Fig. A2. The spatial-averaged basal melt is relatively small in the first two years, increasing rapidly by several orders of magnitude over the next 13 years, and then transitions to a dynamic steady state with a period repeating response, we refer to as a basal melt rate oscillation, with the melt peak occurring around six years after the first three decades. Over the last 70 years, the basal melt rate (3.56 m yr$^{-1}$) oscillations had an average amplitude (peak-to-trough) of $\sim 0.7$m yr$^{-1}$. We tracked the changes of melt rate of four ocean cells at different locations along the centre line of domain (yellow line in Fig. 2). Fig.
A3 indicates that melt rates of cells close to the GL will experience a rapid increase in the first few years since it become wet, and then decrease with an oscillation pattern but decreased amplitudes until the cell is far away from the new grounding zone. The green and blue points experience rapid increase in melting in the first few years but are largely different in magnitude (Fig. A3b). The green point experiences the initial spin-up and subsequent slow increase in melting during the intrusion of the

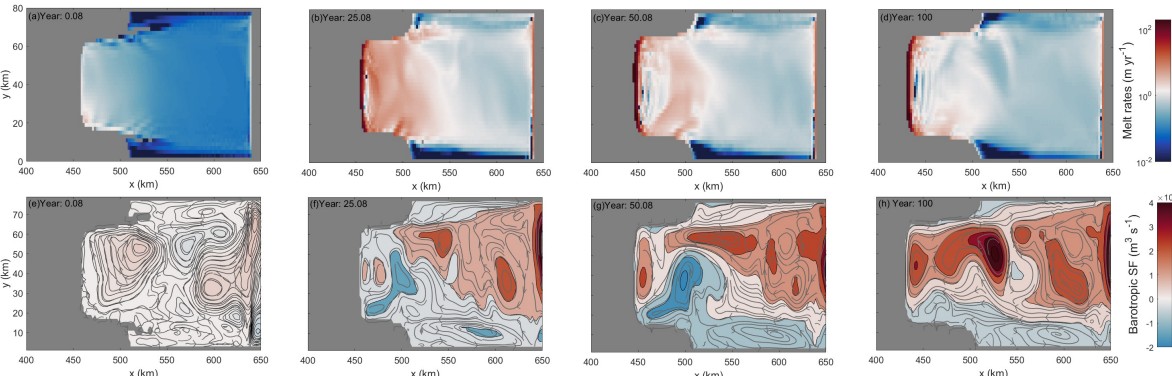

**Figure 4.** Spatial pattern of basal melting (m/yr) (top row) and barotropic streamfunction XY sections $(m^3/s)$ with circulation contours extracted from the barotropic flow field (bottom row) at different years from CTRL.

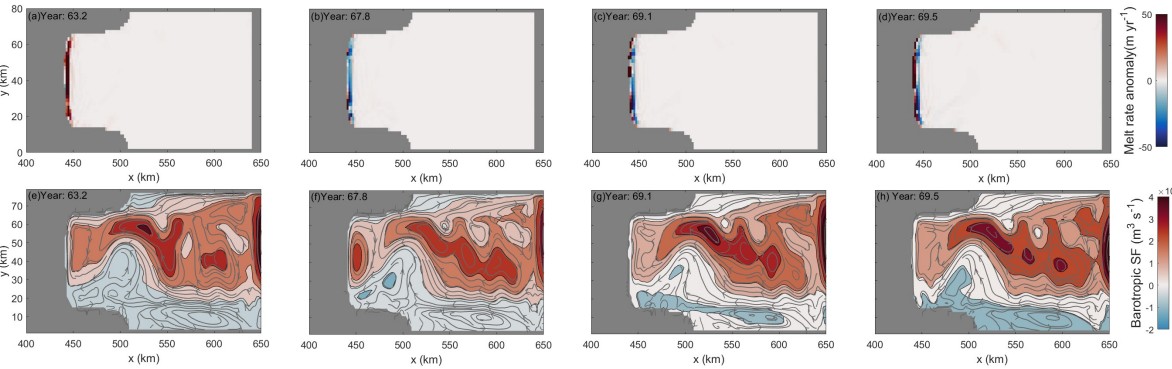

**Figure 5.** Spatial pattern of basal melting anomalies with respect to the mean melt rates over the oscillation cycle period (m/yr) (top row) and barotropic streamfunction XY sections $(m^3/s)$ with circulation contours extracted from the barotropic flow field (bottom row) around one oscillation cycle from CTRL. The corresponding time points are shown in Fig. 3.

warm water from the northern boundary, while the blue point experiences much higher melting immediately since becoming
ungrounded. It suggests that the high melt rates occur near the GL within a short distance less than 6 km.

Spatial distribution of basal melting at different years (top row of Fig. 4) indicates a high melt zone near the GL. The ocean circulation in the ice shelf cavity is characterised by inflow on the east and outflow on the west (bottom row of Fig. 4) with periodic decay of the circulation strength that coincides with the melt rate oscillations (bottom row of Fig. 5).

Examination of the results over the last 70 years suggests that three main processes could drive the basal melt rate oscillation
and upon initial inspection appear to be correlated with each other. The three main processes include 1) the discretisation of GL retreat, 2) ocean dynamics such as eddies and gyres, or 3) changes of ice shelf geometry that might have a feedback between changes to the ocean model layering and the ocean circulation solution, which will discussed in Sec. 4.1). The"discrete

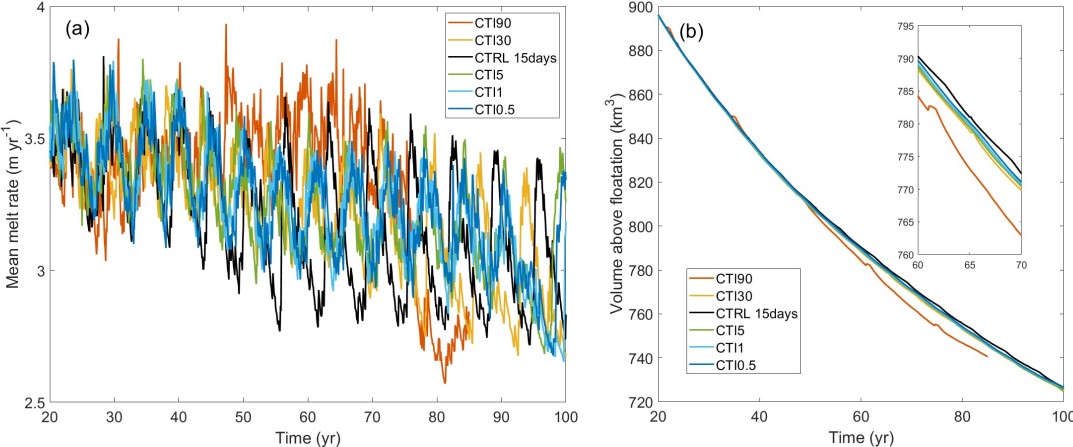

**Figure 6.** (a) Simulated mean melt rates and (b) ice volume above floatation with different coupling interval. The inset box in (b) is the zoomed in period between year 60 to year 70.

GL retreat" refers to the GL retreat occurs over a row of ocean model cells at a time, due to discritization of the models approximately aligned with the GL.

We present two parameters to ascertain the influence of these processes on the melt rate oscillation response. These are 1) the rate of ungrounding calculated as moving changes in grounded area with a time window of 12 months, and 2) the gyre circulation within the ice shelf cavity calculated as the strength of the barotropic streamfunction. To quantify if a correlation between the basal melting and the rate of ungrounding and gyre circulation exists, we calculated the correlation coefficients with different time lags (see internal panel in Fig. 3).

The highest correlation coefficient between basal melting and GL retreat (Fig. 3a) is 0.73 with basal melting lagging GL retreat by 5 months. The relationship between the row index of GL and the basal melt indicates that the melt peak comes 5 months later than a new row of GL (see Fig. A5). It suggests that the discrete ungrounding is highly correlated with the melt peak, which is confirmed through examination of the finer 1-day outputs (see Fig. A4). There is a high correlation (0.99) with no lag between the gyre circulation and basal melting (see Fig. 3b).

**3.2 Coupling interval influence**

The coupling interval is the time interval between exchange of data between ice and ocean models, which does not vary within a given simulation. To explore the influence of coupling interval on simulating ice-ocean interaction, we performed additional sensitivity experiments with different coupling interval as shown in Table 1. The simulated mean melt rates (Fig. 6a) and the ice volume above floatation (Fig. 6b) indicate very little sensitivity to the coupling interval between 0.5 days and 3 months

in the general trend. This is consistent with sensitivity tests with coupling periods ranging between 1 month and 1 year using NEMO-Elmer/Ice (Favier et al., 2019), in which the mean cavity melt rate seen by Elmer/Ice shows very little sensitivity to the

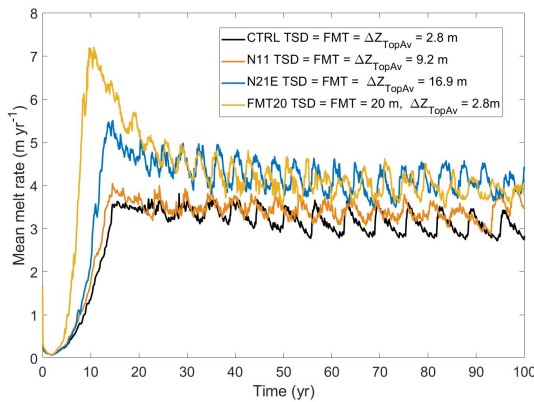

**Figure 7.** Simulated mean melt rates with different vertical resolutions (spatially-averaged top layer thickness $\Delta Z_{TopAv}$) and the imposed depths for mixing fluxes (flux mixing thickness; FMT) and sampling tracers (tracer sampling distance; TSD).

coupling period. However, CDT90 does not show an obvious oscillation pattern compared with the other experiments, which implies that using a coarse coupling interval may lead to the loss of temporal detail in the coupled ice sheet/ocean response. It can also be seen in the tests with 6-month and 12-month coupling periods in Favier et al. (2019), in which the oscillation

feature was obviously smoothed. Additionally, mild variations in periodicity and magnitudes are found as the coupling interval varies. Tests with coupling interval of 5 days or less show more consistency, while tests with coupling intervals of 15, 30, 90 days show differences in magnitudes and phases. CDT30 is closer than CTRL (15 days) to the shorter coupling intervals, suggesting that there might be some cancelling effects in CDT30. Further study to understand the causes and nature of the impact of coupling intervals greater than 5 days would be of benefit to the coupled ice - ocean modelling community.

**3.3 Ocean model vertical resolution influence**

Vertical discretisation and therefore the resolution of the ocean model near the ice/ocean boundary, are known to have a significant effect on the basal melting through differences in the distribution of meltwater fluxes and the calculation of thermal driving (Gwyther et al., 2020). To explore the effect of ocean model vertical resolution in the coupled system, we ran the model with different vertical resolutions as shown in Fig. 7. The geometry and chosen vertical scaling coordinate in CTRL (Fig. A6)

produced top-layer cells of thickness equal to 0.16 m adjacent to the GL (0.52 m in N11, 0.95 m in N21E), 3.09 m at mid-ice shelf (10.23 m in N11, 18.78 m in N21E), and 4.10 m at the ice front (13.68 m in N11, 25.35 m in N21E).

  As expected, experiment N21E (with the coarsest ice–ocean boundary layer), produced the highest melt rates as compared with the CTRL and N11 experiments (Fig. 7). The result confirms that coarser vertical resolution in ice shelf cavities may overestimate the melt rates due to generating stronger implicit vertical mixing (Gwyther et al., 2020). A finer resolution of the

top model layer region in CTRL allows a better representation of the thin meltwater layer, increased stratification and better insulation of ice from water below (Gwyther et al., 2020).

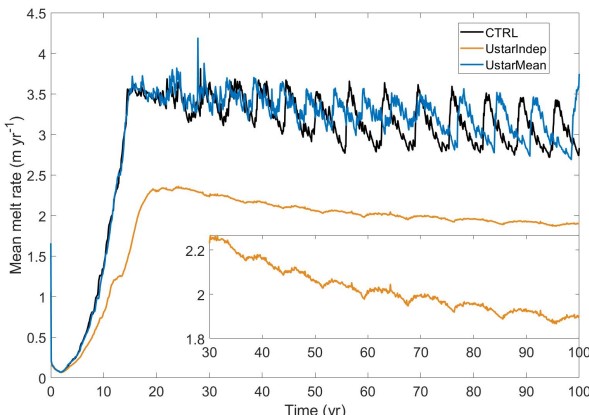

**Figure 8.** Simulated mean melt rates from experiments CTRL, UstarMean, and UstarIndep. The inset panel is the zoom of experiment UstarIndep in the y axis.

In Gwyther et al. (2020), modified mixing experiments using ROMS adopted a vertical mixing scheme similar to the Losch-style scheme (Losch, 2008) by imposing prescribed tracer sampling distance (TSD) and flux mixing thickness (FMT) of 20 m in z-level models, which displayed higher melting than the standard ROMS configuration. In the coupled system, a similar test FMT20 was done and comparison with CTRL (Fig. 7) confirmed the higher melt rates in FMT20 due to higher vertical heat transfer from below into the top model layer. A similar oscillation pattern existed in all of the experiments related with vertical resolution, but showed different frequencies and amplitudes. The outcomes of these experiments demonstrate that emergence of the basal melt oscillation does not depend on the vertical resolution of the ocean model.

However, the implications of the vertical resolution dependence on melt rates for the coupled model system are that the melt rates have a small but influential numerical dependence between melt rates and ice shelf geometry changes. With an ice shelf that experiences thinning and retreat, the ocean model response is to develop thicker layers which will therefore produce higher melt rates, and vice versa for an ice shelf that thickens. This feedback will amplify any response to the ice sheet where a thinning ice shelf, for example, may therefore experience accelerated rates of higher melt rates leading to enhanced thinning. The feedback response may be particularly important near the grounding zone where cells that were dry and masked to produced melt rates may evolve the fastest thickness changes as the wet cells continue to inflate, therefore leading to melt rates that become artificially higher over time.

### 3.4 Dependence on the friction velocity

The friction velocity, $u_*$, determines the surface shear stress that drives the turbulence which mixes heat and salt from the ocean below to the ice base, and hence directly affects the basal melting.

In the CTRL simulation (and all simulations in which $u_*$ is set to TOP in Table 1), the friction velocity is calculated from the water velocity in the top model cell.

To explore the impact of the way we calculate $u_*$ on the basal melting, we undertook two additional experiments: 1) UstarMean, in which the mean velocity from the top three ocean model layers is used to calculate $u_*$, and 2) UstarIndep, in which we used constant values of thermal and salinity exchange velocities at the ice-ocean interface ($\gamma_T = 1 \times 10^{-4} ms^{-1}$,

$\gamma_S = 5.05 \times 10^{-7} ms^{-1}$). The chosen values match those used by Hellmer and Olbers (1989), and are approximately equivalent to a constant friction velocity of $0.01 \ ms^{-1}$.

Results show that basal melting is relatively insensitive to the dependency of friction velocity on the vertical resolution (comparing CTRL and UstarMean in Fig. 8). The shift of melt peak between CTRL and UstarMean may be due to the different timing of GL movement in both experiments. Experiment UstarIndep shows generally lower melt rates, likely in accordance

with a choice of a somewhat smaller value for the fixed transfer coefficients. More noteworthy, the simulation still shows the oscillation pattern but with a short time lag of 1 month (not shown here) and a much smaller amplitude compared to the simulations with velocity dependent transfer coefficients. This indicates the existence of a feedback mechanism, in which the periodic melt-driven acceleration of the flow further enhances the heat transfer to the ice, which increases the overall magnitude of the oscillations.

### 3.5    Preservation of tracer properties

The layer thickness of the ocean model evolves during the simulations through sea surface height changes due to ocean dynamics and ice shelf thickness changes. In the ocean model by default, the non-conservative nature of volume integrated tracer properties is negligible due to relatively small vertical transformations, resulting in minor changes of total integrated temperature and salt content in the ocean domain. However, in the coupled system, the rate of change of the individual layers and total

water column thickness is much larger than sea surface height changes due to ocean dynamics. In the case of large variations in water column thickness, conserving volume integrated properties through ice draft change can be expected to introduce a non-physical drift in tracer properties. We consider the alternative approach that preserves absolute tracer values through water column thickness adjustments due to ice draft evolution. This preservation of absolute tracer values through ice draft adjustments is implemented in our CTRL experiment, while, for comparison, conserving volume integrated tracer properties

is considered in our PIT experiment. Note that the handling of tracer properties through ice draft change is separate from the way in which basal melting is implemented, and the latter is imposed on the ocean model through salt/heat fluxes Galton-Fenzi et al. (2012). In response to the ice draft change, we simply change the volume of the water column without adding any fluxes.

Fig. 9a indicates significant difference in the basal melt between the default and our alternative approach. Experiment PIT provided generally higher mean basal melts with some abnormal oscillation and higher amplitudes, which was quite different

from the CTRL experiment. Fig. 9b and c show the temperature - salinity (T-S) distribution at a given point in time for both simulations relative to a meltwater ocean mixing line (McDougall et al., 2014; Gade, 1979). The meltwater ocean mixing line indicates the viable evolution of T-S properties purely in response to melting/freezing, hence any deviation from the mixing line must be due to physical mechanisms other than melting/freezing in order to be considered plausible. Figure 9c presents a strong drift of the tracer values towards the colder/fresher and colder/denser conditions while the volume integral of

properties is conserved in the PIT experiment. The conservation of absolute tracer properties in CTRL (Fig. 9b) shows much

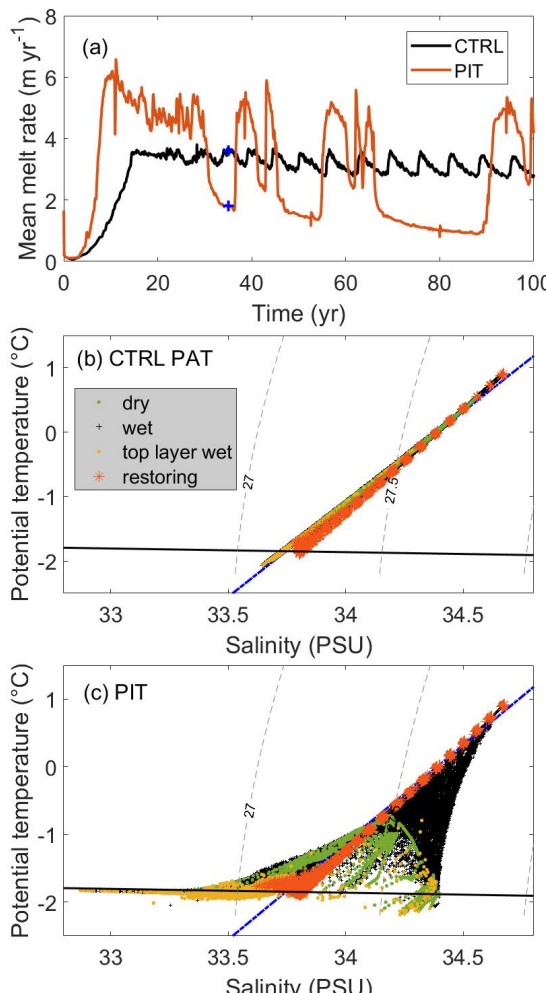

**Figure 9.** (a) Mean basal melting from experiments CTRL and PIT. The salinity-temperature diagram of (b) CTRL and (c) PIT at year 35 (blue cross points in a). The black line is the surface freezing temperature, the blue line shows the meltwater ocean mixing line, and the grey dashed lines show the potential density anomaly contours ($kg/m^3$) according to labels

more reasonable tracer evolution following the mixing line, supporting choice of this approach over conservation of integrated properties, although the total ocean heat and salt contents are not conserved in this approach.

### 3.6 Tracer properties of dry cells

Due to the relatively small changes in tracer values in the ocean that can induce circulation changes and therefore feedback on
the melt rate, the tracer values prescribed to the dry cells that become exposed to the ocean during ungrounding are an important

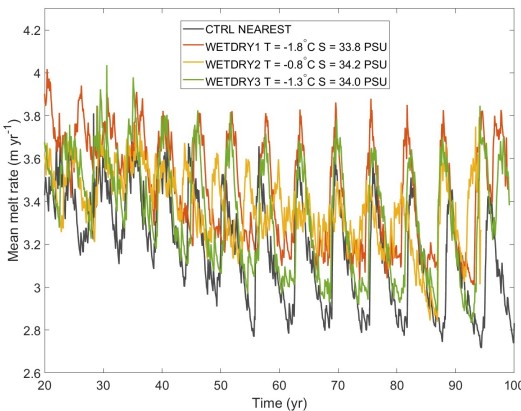

**Figure 10.** Simulated mean melt rates with different tracer properties for the nudging dry cells.

consideration. To explore the influence of the assigned dry cell tracer values, as an alternative to our CTRL experiment (in which the tracer properties in dry cells are assigned the values of the nearest wet cell), we conducted a series of experiments in which fixed tracer properties are imposed in dry cells as restoring targets with rapid restoring rates. The restoring method is the same as the forcing at open ocean end of the domain with a restoring rate of 10 $days^{-1}$. The FVCOM-Elmer/Ice group used a similar wetting and drying scheme with fixed tracer properties applied on the dry cells immediately when the cell turns from dry to wet (Zhou, personal communication).

Here we present the results of three additional experiments to represent dry tracer cells comprised of 1) cold (T = -1.8 °C) and fresh (S = 33.8 PSU, density = 1027.23 $kg/m^3$) water (WETDRY1), 2) warm (T = -0.8 °C) and salty (S = 34.2 PSU, density = 1027.52 $kg/m^3$) water (WETDRY2), and 3) a compromise between the two (T = -1.3 °C and S = 34.0 PSU, density = 1027.38 $kg/m^3$) (WETDRY3). A similar oscillation pattern can be seen in all these experiments with different amplitudes and frequencies (Fig. 10 showing the period between 20-100 years and Fig. A7 with the initial spin-up stage included), which suggests low sensitivity of melt oscillations to the initialisation of dry cell tracer values. However, initialisation of dry cell tracer values does have a non-trivial impact on melt rates. In general trend, WETDRY1 with cold and fresh water provides highest melt while CTRL shows lowest melt rate. WETDRY3 shows similar amplitudes and periods in oscillation with CTRL while WETDRY2 with warmer water presents smallest amplitudes except for the last two decades. It indicates the oscillation feature can not be removed by using different initialisations of dry cells but the initialisation of dry cells would impact the oscillation pattern with different degrees.

## 4 Discussion

Performing simulations in a coupled ice-shelf/ocean system with an evolving ice draft and GL introduces several complexities that must be considered for robust projections of future ice sheet evolution in response to ocean climate change. Assessment

of parameterising sub-shelf melt rates in an idealised coupled system is significant and necessary for the benefit of application on a real world.

## 4.1 Melt oscillations

We have demonstrated that the oscillations in mean melt rates are directly associated to the discrete retreat of the GL in our structured grid ocean model ROMS (Sec. 3.1). We now consider whether such oscillations feature in comparable models, and what factors may cause or enhance the oscillations.

We compared the simulated mean melt rate from CTRL with those from other coupled models participating in the MIS-OMIP1 project with a common set of parameters (the COM configurations) (Fig. 1, pers. comm. Xylar Asay-Davis). All the contributing ocean models used the same horizontal resolution of 2 km while the ice modes used different horizontal resolution near the grounding line ranging from 200 m to 1 km. A large range of post-spin up behaviours is seen. NEMO-UKESM1is displays a qualitatively similar behaviour to ROMS-ElmerIce (CTRL in the current study). Several simulations display some variability, but without such clear periodic oscillations. FVCOM-ElmerIce gives the most stable melt, with very little change after the initial spinup period.

The cause of the melt variability, whether oscillatory or otherwise, has not been clearly established in these models, though correlations with discrete ungrounding and the strength of gyres in the ocean model have been demonstrated in the current study (see Sec. 3.1 and Fig. 3). Given this correlation, and the fact that FVCOM uses an unstructured mesh of triangular elements, it is perhaps not surprising that FVCOM-ElmerIce does not display the melt oscillations.

The melt oscillations did not emerge in the stand-alone ocean simulations Ocean1 or Ocean2 from the Ice Shelf - Ocean Model Intercomparison Project + (ISOMIP+; Asay-Davis et al. (2016)), which used a fixed topography. The oscillations do occur in our ISOMIP+ Ocean3 simulation (with relatively low amplitude), a stand-alone ocean simulation in which a evolving ice geometry is prescribed annually (Fig. 3).

The fact that they occur only in simulations in which the GL moves, together with the close relation between GL retreat and mean melt, strongly suggests that the melt oscillations are driven by the discretised ungrounding that occurs on a structured grid that is aligned with the GL. The grid orientation and the experiment design in this study guarantee the central part of the GL aligned with the grid, which allows the ungrounding of a whole row of grid cells to occur approximately together. A grid rotated to about 45 degrees would potentially allow a different pattern of ungrounding to appear. If the grid were rotated to about 45 degrees, the experiment design would not encourage the ungrounding of a whole row of cells and cells may unground one at a time instead. We do not know whether the melt oscillations would then occur the same as in the current set up, with reduced strength, or not at all. Reduced strength seems most likely since smaller scale discrete ungrounding would still occur. A further test with a rotated grid in the ocean model might help to diagnose the potential numerical issues associated with coupled grounding line retreat processes. Additionally, We don't know how many new cells are needed to be exposed to the ocean to cause this instability. Using a finer horizontal resolution in the ocean model may solve this problem and further study is needed to explore how finer will be enough. Fig. A8 shows the simulated mean basal melting from contributing coupled models using both COM configuration and typical parameters (TYP configuration) with different horizontal resolution in ocean models.

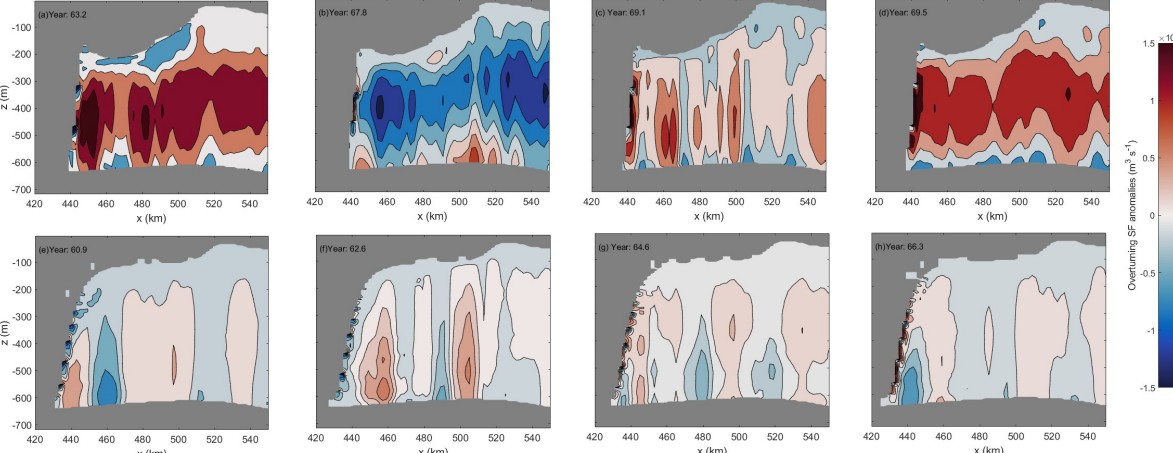

**Figure 11.** XZ sections of anomalies of overturning streamfunction ($m^3/s$) near the grounding line from CTRL (top row) and Ocean3 (bottom row) around one oscillation cycle. Anomalies are calculated with respect to the whole cycle. The chosen time points are shown with red points in Fig. 3.

There are no obvious oscillation features in both COM and TYP simulations using NEMO-ElmerIce but the noise has been smoothed in TYP with a finer ocean resolution (1 km). The basal melting pattern from NEMO-UKESE1is-COM is pretty similar with CTRL in this study while the TYP simulation (light blue line, ocean horizontal resolution of 8 km) shows much worse oscillations. However it is hard to say that the coarser ocean horizontal resolution led to the worse oscillations since there are some other different configurations between their COM and TYP configurations. It is still difficult to discriminate whether

the oscillation feature is a fundamental emergent intrinsic physical feature of the coupled system, or a numerical artefact that arises through the combination of multiple processes.

  The buoyancy driven overturning circulation under the ice shelf (Fig. 11) can be strengthened by a positive feedback involving basal melt contributing to increased buoyancy forcing, which drives a faster and divergent flow that can both entrain more far-field ocean water and increase friction at the ice-ocean interface that can both lead to increasing melt. The positive

buoyancy-melt feedback is represented in our simulations by the dependency of basal melt on friction velocity, $u_*$. The fact that experiment UStarIndep shows small amplitude melt oscillations suggests that the buoyancy-melt feedback enhances the magnitude of melt oscillations seen in most of our simulations. A shorter time lag in UstarIndep confirms that the buoyancy-melt feedback is delayed due to the dependency of basal melt on $u_*$.

  Another possibility is the evolving slope of the base of the ice shelf and the melt oscillations. Melt rates peak in the zone of

maximum slope (hereafter referred to as 'steep shelf zone') (top row of Figs. 4, 5), due to the relationship between shelf slope and speed associated with the pressure gradient. Both slope and extent of this zone can therefore significantly impact on melt rates, through the buoyancy-melt feedback, where a steeper slope can lead to an accelerating flow and therefore a higher melt rate and vice versa. This ice shelf geometry can potentially vary with discrete ungrounding for two reasons, both related to the

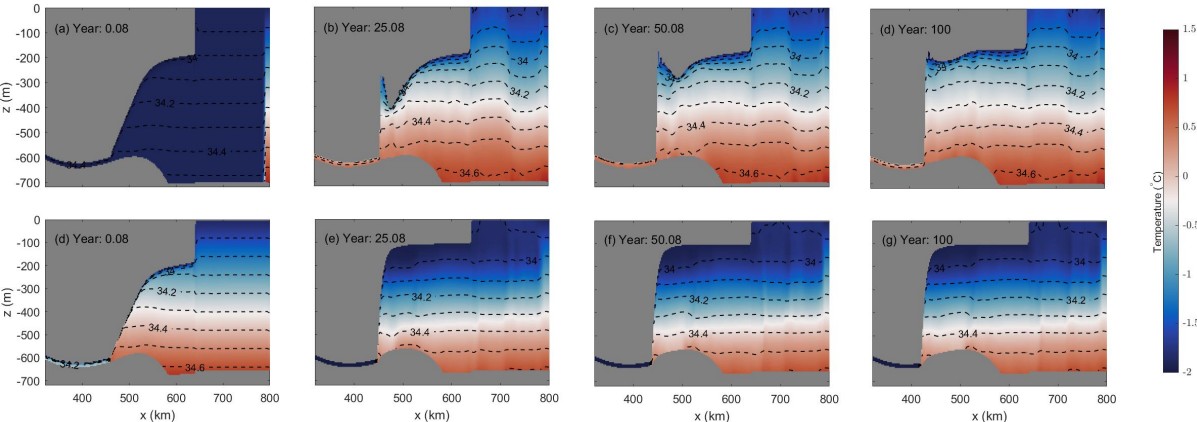

**Figure 12.** Ocean temperature (°C) and salinity (PSU) sections through the centre line of the domain (white dashed line in Fig. 2b) from CTRL (top row) and Ocean3 (bottom row).

fact that the steep shelf zone is adjacent to the GL and occurs over ∼3 grid cells after the initial spin-up (see Fig. 12). This
scale and proximity mean that, firstly, a single row of ungrounding can significantly increase the relative size of the steep shelf zone, and, secondly, that a single row of ungrounding can significantly alter the mean slope in this region. The trapped narrow band of circulating ocean water (Fig. 12) due to the icemount geometry feature near the grounding zone in CTRL (see 12b to d for ice shelf geometry evolution) may have also enhanced the oscillation compared with Ocean3.

Comparison of the melt oscillation's amplitude and correlation lag between Ocean3 and CTRL may be informative to further
improving understanding. Ocean3 demonstrates small amplitude melt oscillations with no lag, whereas CTRL demonstrates high amplitude melt oscillations that lag the GL retreat by approximately 5 months. The simplest consistent explanation for these behaviours is that GL retreat initially triggers a small but instantaneous response of the melt rate that would be expected, in both simulations. A larger delayed (5 month) response occurs in the melt rate from the coupled system, over a coupling interval (15 days), due to geometry changes as previously described.

Here is our understanding of the hypothesised mechanism behind each cycle: The mean flow develops in response to the ice shelf melting (see Fig. 5e) whereby a single clockwise gyre under the entirety of the ice shelf forms in response to the enhanced melt-buoyancy-driven circulation feedback. The high melt phase of the cycle occurs in response to accelerating flow and the transport of warmer waters to the deeper part of the ice shelf driving higher melt rates is correlated with ungrounding of the ice shelf. At the point of maximum melting, and associated thinning of the ice shelf, the single gyre is reorganised
and a smaller circulation feature intensifies adjacent to the deep grounding line (see Fig. 5f). The reduction of the circulation during the low melt rate phase is due to a both cooler and slower ocean, and also the lack of intrusion of relatively warm waters. However, the minimal amount of melting is sufficient to again start to drive a buoyancy driven flow that over time re-establishes the single gyre circulation feature that is again strengthened due to further ungrounding, and the process repeats. Our understanding of what leads to the development of the flow reorganisation is unclear and may be due to either a direct

physical mechanism or numerical artefacts. Although the sensitivity of the amplitude of the melt oscillations to the friction velocity (see Sec. 3.4) suggests that there is a critical feedback between melting and circulation we are unable to determine the exact cause. Further investigation is needed to understand the relative influences of both numerical and physical processes that arise from the coupled system.

In summary, we hypothesise that, while the melt oscillations are triggered by discretised ungrounding, they are greatly
enhanced by a combination of steep slopes in ice draft near the GL, the tight coupling of ice draft and ocean circulation evolution, and the buoyancy-melt feedback, which are physically plausible features. Our results however also suggest that the pattern of ungrounding is controlled by the discretisation of the coupled system (primarily the ocean grid) and future work should investigate the use of a grid rotated to about 45 degrees to test the sensitivity. In a real-world simulation, in which the GL is not aligned with the model grid, do these melt oscillations still occur in the similar way? We also recommend future
studies by employing finer resolution near the GL in the ocean model and quantifying the impacts of finer resolution and grid rotation to determine whether the time-mean melt in the current study is affected by numerical artefact.

## 4.2 Ocean tracer properties in a coupled model

We have considered the implications for tracer properties of how to handle evolution of the ice draft (Sec. 3.5), choice of vertical advection scheme (discussed below), and using a "wet-dry" (or "thin film") scheme for GL evolution (Sec. 3.6).
The ice draft evolves in response to both ocean-induced melting and evolving ice dynamics, which in general features non-zero and time-varying horizontal flux divergence. The dynamical ice draft evolution imposes a time-varying pressure on the ocean. However, in the physical system, the ocean circulation adjusts at the barotropic wave speed to pressure forces than the time scales on which ice geometry changes occur. Hence, at time scales relevant for the ocean, the pressure forcing imposed by the time-varying ice draft is neglected and in ROMS, water is added (removed) locally to (from) the water column as needed
to accommodate the geometry change.

The delayed coupling time response therefore requires a choice about how to assign tracer properties when water is added. It is not correct to simply assume the additional water has properties of fresh melt water (S=0 psu), because the volume change in a given water column does not in general balance the ocean-induced melt/refreezing.

The freshening impact of meltwater is, in ROMS and most other ocean models with ice shelf cavities, dealt with by imposing
a salinity flux, which is independent from the volume change due to ice geometry changes. For a static ice geometry, this is equivalent with assuming that any ice thickness changes due to melting or freezing are instantly compensated by the ice dynamics. Our experiments of how to handle preservation of tracer properties in response to ice draft change (Sec. 3.5) suggest that preservation of absolute tracer values is giving plausible T-S evolution in response to changes in water column thickness. However, this approach does not explicitly resolve the partitioning between ice dynamical and melt driven geometry changes,
and it is not expected to maintain conservation of tracer properties across the domain, and future studies would be wise to quantify possible tracer drift.

ROMS offers several choices for handling the vertical advection and diffusion of tracer properties. Having tested several of these options (not shown), the best choice for obtaining a plausible T-S evolution was given by using a conservative, parabolic

splines reconstruction for vertical diffusion on active and passive tracers. Omitting this option leads to the development of non-physical tracer values in shallow and confined regions near the GL of the ice shelf cavity, typically identified by too high temperatures near the seafloor and too low temperatures near the ice base (when compared to the meltwater mixing line (McDougall et al., 2014), ambient water and pressure melting point constraints). Similar behavior has been observed in other contexts using ROMS in static, but more realistic ice shelf cavity configurations, e.g. near the Foundation Ice Stream on Filchner-Ronne Ice Shelf (Daae et al., 2020), as well as in static and time-evolving ice shelf cavity simulations using FVCOM (Zhou and Hattermann, 2020). Within FVCOM, the use of a positive definite MPDATA vertical advection scheme successfully suppresses the development of such non-physical tracer values, and our working hypothesis is that the large vertical gradients that are imposed by the meltwater forcing (and the resultant horizontal divergence in the upper part of the water column with its associated upwelling near the GL) tend to cause overshoots in the vertical advection-diffusion balance. The use of more conservative numerical schemes, as is reported here, may be instructive for other studies, while further investigations might be needed for a more detailed understanding of the cause of these instabilities.

The use of a "wet-dry" scheme is to enable GL movement in the model, and does not directly represent any real world process. The wetting of dry cells as the GL retreats could be considered a proxy for the outflow of subglacial meltwater. However, given that the volume flux in the coupled model is essentially determined by thickness of the dry cells and rate of ungrounding, rather than by subglacial processes such as friction heat and geothermal heat under the grounded ice, this is not quantitatively realistic.

In the current study, we did not find a strong sensitivity of melt oscillations to our choice of how to handle dry cell tracer properties. This suggests the values assigned to the dry cells are not a strong influence on the melt oscillations, but it does affect the melt rates in general trend, magnitudes, and frequencies. Colder and fresher dry cells in WETDRY1 drive more overturning through buoyancy when becoming wet and lead to slightly higher basal melting than other experiments. However, the only combination of approaches to handling vertical advection schemes and ice draft evolution was to 1) preserve absolute tracer values through adding/removing water due to water column thickness changes and 2) use conservative, parabolic splines reconstruction of vertical derivatives to suppress numerical overshoots caused by large vertical gradients near the GL.

## 5   Conclusions

We evaluated the impact of a wide variety of ice-ocean interaction parameterisations on the coupled ice-ocean system through an ensemble of experiments. These experiments were conducted within a new coupling framework (FISOC), which combined an ocean model modified for ice-ocean interaction (ROMSIceShelf) and an ice sheet model (Elmer/Ice). An oscillation pattern in the simulated spatial-averaged basal melting rates is found to be strongly associated with the discrete GL retreat, which is also detected in experiment with a prescribed geometry change (that is, uncoupled from any dynamic ice feedback). We propose that this oscillation feature is triggered by the discretised ungrounding and largely amplified by a combination of physically plausible mechanisms including the steep shelf zone near the GL, the buoyancy-melt feedback, and the tight coupling of ice draft and ocean circulation evolution. A series of sensitivity tests showed that the existence of this oscillation pattern was

insensitive to the choice of coupling interval, vertical resolution of the ocean model, the initialisation of tracer properties of the dry cells, or the dependency of friction velocities to the vertical resolution. Future studies with a higher horizontal resolution and a rotated ocean model grid will help further quantify the impact on this oscillation feature, and determine whether the melt oscillation is a numerical model artefact.

While the existence of the melt oscillations is robust to our various model configurations, we find that our model choices have a non-trivial impact on mean melt and ocean circulation strength, which might be interesting to the coupled ice ocean system community. As the ice draft (and hence water column thickness) evolves, a choice must be made about tracer conservation in the ocean model. Since the model default option to conserve the tracer volume integral introduces spurious tracer drift under a changing ice geometry, we locally conserve absolute tracer values when adjusting the ice draft to obtain a physically plausible tracer evolution. Furthermore, the numerical stability of our experiments with an idealised cavity geometry appears to be sensitive to the choice of the vertical advection/diffusion scheme, where large vertical gradients associated with the melt-water driven upwelling near the GL have a tendency to introduce numerical overshoots.

*Code availability.* The FISOC-ROMSIceShelf-Elmer/Ice source code, version information for related software, and input files needed to run the experiments described in the current study are all publicly available (https://doi.org/10.5281/zenodo.5908713).

## Appendix A: Experiments with different time resolutions in ROMS

To explore the sensitivity of simulated basal melting to the time resolutions in the ocean model, we fixed the ice geometry and ran the ocean model alone with various combinations of barotropic and baroclinic timestep sizes in the ocean model. The designed experiments are shown in Table A1.

**Table A1.** Experiments with different combinations of barotropic and baroclinic timestep sizes. N is the number of barotropic timesteps between each baroclinic time step.

| Simulation | baroclinic DT | N | barotropic DT |
|---|---|---|---|
| DT200N30 | 200 | 30 | 6.67 |
| DT100N15 | 100 | 15 | 6.67 |
| DT200N60 | 200 | 60 | 3.33 |
| DT100N30 | 100 | 30 | 3.33 |
| DT200N120 | 200 | 120 | 1.67 |
| DT100N60 | 100 | 60 | 1.67 |
| DT50N30 | 50 | 30 | 1.67 |

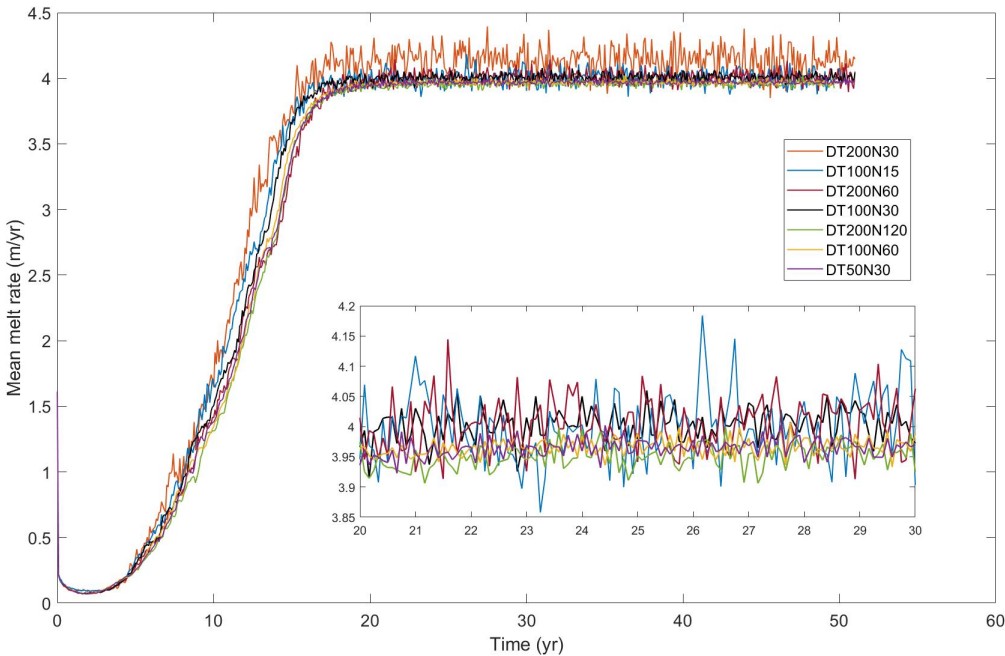

**Figure A1.** Simulated mean melt rates from tests with different combinations of barotropic and baroclinic timestep sizes shown in Table A1

**A1**

*Author contributions.* CZ led experiment design, implementation of experiments, and paper writing. RG, BGF, DG, TH contributed to experiment design. All authors contributed to paper writing and discussion of ideas.

*Competing interests.* The authors declare that they have no conflict of interest.

*Acknowledgements.* Chen Zhao and Ben Galton-Fenzi received grant funding from the Australian Government as part of the Antarctic
Science Collaboration Initiative program (ASCI000002). Rupert Gladstone is supported by Academy of Finland grant number 322430. Tore Hattermann acknowledges financial support from Norwegian Research Council project 280727. This research/project was undertaken with the assistance of resources and services from the National Computational Infrastructure (NCI), which is supported by the Australian Government. We thank the MISOMIP community by sharing their basal melt rates using different coupled models, that provoked this study.

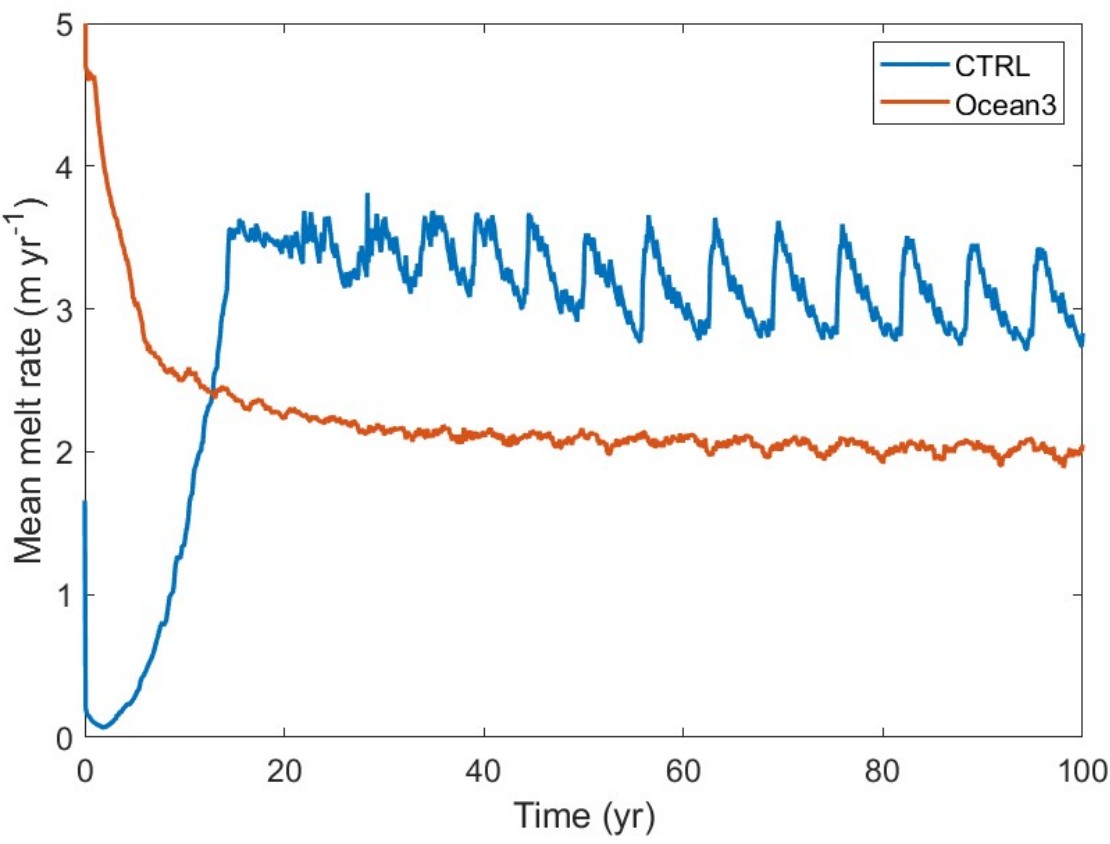

**Figure A2.** Simulated mean melt rate from CTRL and Ocean3.

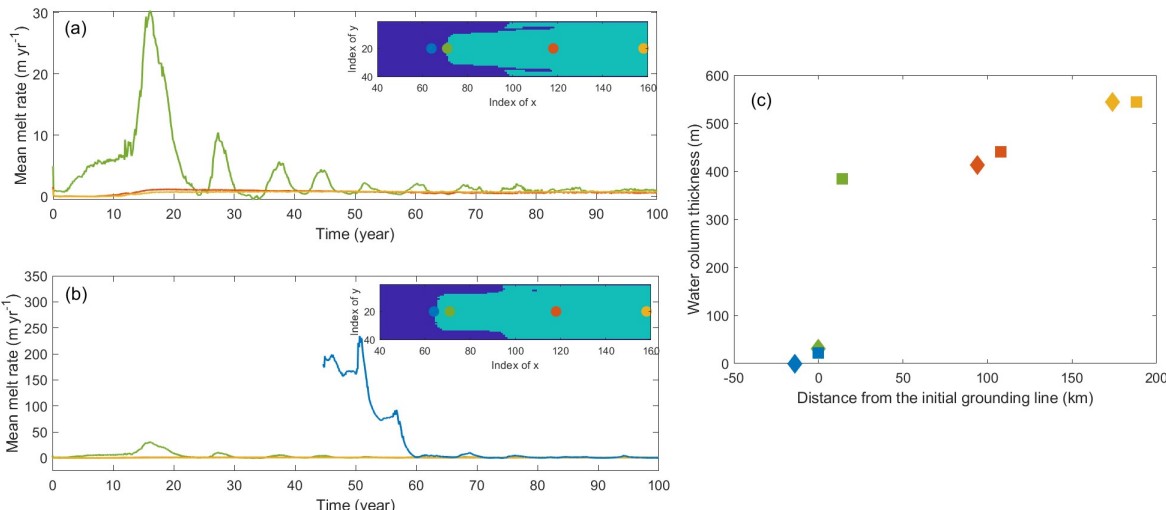

**Figure A3.** Changes of melt rate of ocean cells at different locations along the centre line of domain (yellow line in Fig. 2). The locations of ocean cells are shown in the inset panels with background image indicating the grounded mask of (a) year 0 and (b) year 44. Green and blue dots are located on the grounding line of year 0 and year 44, respectively. The distance of those points from the initial grounding line and the water column thickness are shown in (c). The diamond and square points represent the points in year 0 and 44, respectively.

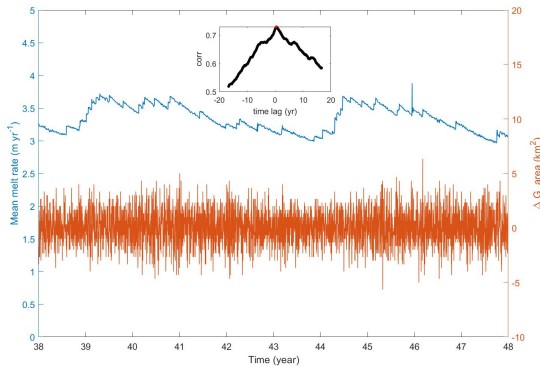

**Figure A4.** Mean melt rate and the moving mean changes in grounded area from the 1-day outputs from year 38 to year 48 of CTRL. The correlation coefficients between the blue line and the orange line are shown in the inset.

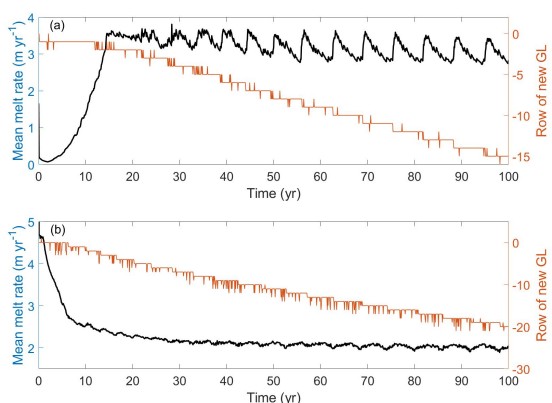

**Figure A5.** Mean melt rate and the row index of new grounding line (GL) from (a) CTRL and (b) Ocean3. The row index of initial GL is 0.

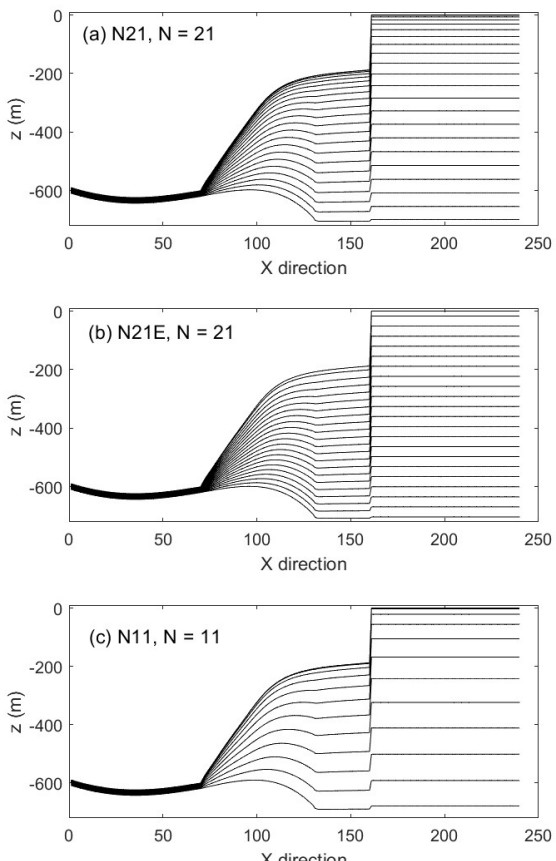

**Figure A6.** Stratification structure of experiment (a) N21 (b) N21E and (c) N11 with different vertical resolution. N is the vertical layers used in the ocean model.

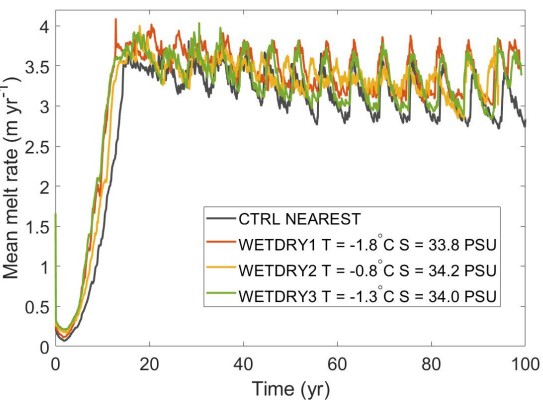

**Figure A7.** Simulated mean melt rates with different tracer properties for the nudging dry cells.

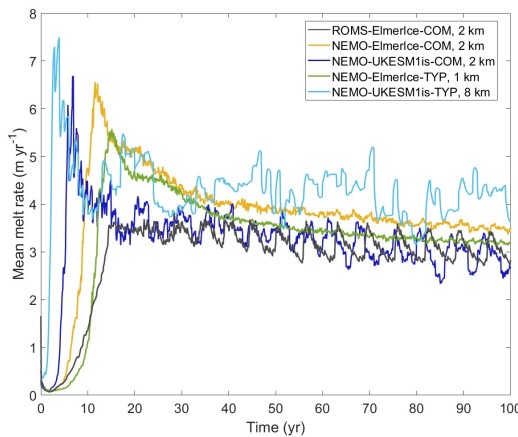

**Figure A8.** Simulated mean melt rates from different coupled models using COM and TYP configurations in MISOMIP1 project (pers. comm. Xylar Asay-Davis).

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
