# Peer review of "Evaluation of an emergent feature of sub-shelf melt oscillations from an idealised coupled ice-sheet/ocean model using FISOC(v1.1)-ROMSIceShelf(v1.0)-Elmer/Ice(v9.0)"

_Geoscientific Model Development, 2022_

## Author Comment (AC1)

**Responses to Reviewer #1**

Review of "Evaluation of an emergent feature of sub-shelf melt oscillations from an idealised coupled ice-sheet/ocean model using FISOC(v1.1)-ROMSIceShelf(v1.0)-Elmer/Ice(v9.0)" by Chen Zhao, Rupert Gladstone, Ben Galton-Fenzi, David Gwyther and Tore Hattermann.

Recommendation: minor revision

Sub-shelf melt oscillations emerge from coupled ocean-ice-sheet simulations of the Marine Ice Sheet Intercomparision Project (MISOMIP), and this paper investigates the causes of these oscillations. This is a useful study for the ocean-ice-sheet modelling community. The paper is well written and the sensitivity tests make sense and are clearly analysed. I only have minor comments and I suggest to accept the manuscript once they have been considered.

We thank the reviewer Dr. Nicolas Jourdain for the time and efforts spent in reviewing this piece of work. The detailed comments are very much appreciated and will be of great help to improve the quality of this study. We will address all points raised below as part of our revisions. Note that all the line numbers and section numbers in blue refer to the modified manuscript.

Specific comments:

Abstract: it should be reminded that there is no external (atmosphere or sea ice) forcing in the MISOMIP experiments. This would help understand that it is somewhat surprising that an ocean oscillation emerges.

Thanks for the suggestion. But we don't think it is necessary. We already describe the oscillation as an emergent feature, which already implies that it is not forced externally. We make it clearer in the Sec. 2.2 (Line 86). "No external forcing is applied at the surface of the open ocean, which means there is no atmospheric or sea-ice fluxes. A "WARM" forcing, as the only forcing, is applied within a 10 km restoring region near the ocean's northern boundary". We choose to leave discussion about external forcing to the main paper and not the abstract.

L. 53, 152 and at other places: I am not a native speaker, but "couple" should probably be "coupled" (or "coupling" for some occurrences).

Thanks for pointing it out. We have corrected all the words "couple" into "coupled" or "coupling".

L. 97: Weddell and Ross cavities are usually classified as cold, not warm.

Thanks for pointing it out. We corrected it into "...with the warm ice shelf cavities in Amundsen and Bellingshausen Seas".

L. 128: "This parameterization" is a bit unclear.

We mean the three-equation parameterisation here and modified it into "The threeequation parameterisation is typically applied between the top model layer and the ice..."

L. 145-146 (and caption of Tab. 1): It is not clear to me what is the difference between

"conserving the volume integrals of tracer values (temperature and salt)" and "preserve the absolute values, (e.g. heat or freshwater)" as, for example, the volume integral of temperature directly gives the heat content when multiplied by pcp. Furthermore, how exactly is imposed the conservation: additional flux at the surface? uniform T,S correction? Without this information, it is difficult to understand section 3.5.

Thanks for the comment. We don't impose a physical flux. For the basal melting we impose salt/heat fluxes on the ocean model (Galton-Fenzi et al., 2012). For the ice draft change we simply change the volume of the water column without adding any fluxes as such. When melting occurs and freshwater should be added, we remove salt. That's why we have this decision about how to handle tracer properties.

To clarify this, we add a couple of lines here "Changes in water column thickness due to thinning would be maintained through increased ice shelf horizontal convergence/divergence in the ocean circulation in response to mass/volume changes. ROMS effectively introduces a source/sink term imposed by adding or removing heat or salt at the ice/ocean boundary. For example, when the ice shelf melts, the model removes salt/heat rather than adding freshwater volume. The circulation change in this case is a result from density changes rather than volume changes. The approach using a source/sink term of heat/salt transfer imposes a choice upon the ocean model: either conserving the volume integrals of tracer values (temperature and salt) or preserving the absolute values, (e.g., heat or freshwater). Here we will explore the effect of both options on the ocean circulation in a coupled system in Sec. 3.5".

We also added a sentence at the end of Sec. 3.5 "Note that the handling of tracer properties through ice draft change is separate from the way in which basal melting is implemented, and the latter is imposed on the ocean model through salt/heat fluxes (Galton-Fenzi et al., 2012). In response to the ice draft change, we simply change the volume of the water column without adding any fluxes."

Please provide more details on the CTRL and Ocean3 experiments somewhere in section 2 or 3.1 (initial state, temperature and salinity restoring near the northern boundary, coupled models or ocean model with ice draft evolution, etc).

Thanks for the comment. We added one section "Sec. 2.2 Experiment design" for more details about MISOMIP1 and Ocean3.

"Each coupled model experiment in this study was run for 100 years, following Experiment IceOcean1r of MISOMIP1 (Asay-Davis et al., 2016). Like in IceOcean1r, experiments in this study does not include a dynamic calving, in which ice thickness is allowed to be zero without calving. Various configuration in each experiment can be seen in Table 1 and corresponding sections in Sec. 3.

We build our coupled model following the ISOMIP+ projects for stand-alone ocean models with ice-shelf cavities and the MISMIP+ projects for ice sheet models. Result of ISOMIP+ Ocean3 from Asay-Davis et al. (2016) using the same ocean model will be used as a comparison to the control experiment in this study (CTRL in Table 1).

The ocean model in the coupled system is initialised with a steady-state ice geometry from the ice sheet model and a ``COLD" initial condition following Asay-Davis et al. (2016). No external forcing is applied at the surface of the open ocean, which means there is no atmospheric or sea-ice fluxes. A ``WARM" forcing, as the only forcing, is applied within a 10 km restoring region near the ocean's northern boundary (yellow area in Fig. 2a), which is consistent with the warm ice shelf cavities in Amundsen and Bellingshausen Seas. The warm water is expected to reach the ice-shelf cavity within the first two decades and induce strong basal melting and subsequent rapid GL retreat.

In Ocean3, the stand-alone ocean model uses the same steady-state ice topography with the initial state of the coupled system, and is run for 100 years with an annually prescribed evolving ice geometry. The ocean is initialized with the WARM profiles, forced with the WARM profile in the same restoring region with CTRL and strong melting is expected to begin immediately as the sub-shelf circulation spins up. More details about MISMIP+ and ISOMIP+ can be seen in Asay-Davis et al. (2016)."

Fig. 2: is the maximum of the barotropic stream function calculated under the ice shelf or all over the MISOMIP domain?

We calculated the maximum of the barotropic stream function under the whole MISOMIP domain. To make it clearer, we modified the related text into "The highest correlation coefficient between the basal melting and the maximum of the barotropic stream function under the whole domain (Fig. 3b) is 0.99 without a lag within both the 30 days and 1 day outputs". Note the previous Fig. 2 is now Fig. 3 in the revised draft.

Fig. 4: it would be easier to see the signal if the plots were showing anomalies with respect to the mean between year 63 and year 70.

Thanks for the suggestion. But we don't think it is necessary. The significant difference in basal melting across one cycle only occurred in one or two rows of cells where the GL retreated. That's why it looked nearly the same for the basal melting. We modified the color scale to make it look better. See the new figure below.

L. 185-186: what gyre are the authors referring to? Are these the gyres near the northern boundary or the gyre circulation within the ice shelf cavity?

We mean the gyre circulation within the ice shelf cavity. To make it clearer, we modified it into "2) the gyre circulation within the ice shelf cavity calculated as the strength of the barotropic streamfunction."

L. 193-195: I do not understand what the authors want to show with the barotropic circulation: any melt variation is associated with a change in barotropic circulation due to the modified horizontal density gradient and its role in the geostrophic balance (see Jourdain et al., JGR, 2017).

Thanks for the comments. We want to say that the basal melting is very much correlated with the process that we already highly suspected is driving the melt. We modified this sentence into "There is a high correlation (0.99) with no lag between the gyre circulation and basal melting (see Fig. 3b)".

L. 199-204: it is not so much the melt rate that is insensitive to the coupling period (it is actually smoothed for 6-mont h and 12-month coupling periods in Favier et al. 2019), it is the ice-sheet dynamics. Fig. 5 should therefore include another panel to show the ice sheet response (e.g. volume above floatation).

The only way the ocean impacts on the ice dynamics is through basal melting. So if melting is consistent across runs it is reasonable to assume ice dynamic behavior will be too. In this sentence, we made a statement about the sensitivity of general trend in basal melting to the choice of coupling interval rather than talking about the oscillation features. After this statement, we mentioned that CDT90 shows a smoothed oscillation pattern. To make it clearer, we added another sentence at the end "The simulated mean melt rates (Fig. 6a) and the ice volume above floatation (Fig. 6ab) indicate very little sensitivity to the coupling interval between 0.5 days and 3 months in the general trend. This is consistent with sensitivity tests with coupling periods ranging between 1 month and 1 year using NEMO-Elmer/Ice (Favier et al., 2019), in which the mean cavity melt rate seen by Elmer/Ice shows very little sensitivity to the coupling period. However, experiment CDT90 does not show an obvious oscillation pattern compared with the other experiments, which implies that using a coarse coupling interval may lead to the loss of temporal detail in the coupled ice sheet/ocean response. It can also be seen in the tests with 6-month and 12-month coupling periods in Favier et al. (2019), in which the oscillation feature was obviously smoothed. Additionally, mild variations in periodicity and magnitudes are found as the coupling interval varies. Tests with coupling interval of 5 days or less show more consistency, while tests with coupling intervals of 15, 30, 90 days show differences in magnitudes and phases. CDT30 is closer than CTRL (15 days) to the shorter coupling intervals, suggesting that there might be some cancelling effects in CDT30. Further study to understand the causes and nature of the impact of coupling intervals greater than 5 days would be of benefit to the coupled ice ocean modelling community."

We added another panel to show the ice sheet response and see new Fig. 6 below.

---

## Author Comment (AC3)

**Responses to Reviewer #2**

**Summary**

This paper describes the investigation of an interesting phenomenon seen in simulations carried out according to the protocol of the MISOMIP1 intercomparison project. Although only one model is used here, the phenomenon seems to occur in a number of the models that contributed to the MIP, and the factors investigated are technical ones common to many coupled ocean-ice systems, so the paper will be directly relevant to a number of groups working in this field. A such, it's a useful contribution and fits the scope of GMD well. It's generally clearly written and at an appropriate depth, and proof-reading for grammar aside, could be published basically as is.

**General comments**

In terms of the factors that have been tested the work is quite comprehensive, but because the oscillations are present in all the configurations tested and no clear sequence of physical or numerical features that lead to them can be isolated, the conclusions that are drawn in current framing of the paper (why do the oscillations arise) are rather limited. However, for me the most useful thing about this paper is the range of choices of coupling interval, resolution, timestepping etc that are tested and the effects of these choices that are shown up in the changes of amplitude and periodicity of the oscillation. Showing the impact of these choices  - using the oscilliatory behaviour as simply a useful test bed, rather than the main feature of the paper in itself - to others who are configuring their own ice-ocean coupled models is valuable I think, and I would like to have seen more made of these smaller effects, rather than simply stating that the existence of the oscillation is insensitive to the choice.  It would be ideal to have the influence of each factor really broken down into numerical and physical factors, but that seems like an unreasonably huge amount of work, and I think the level of detail that is provided is still useful. Just frame what has already been said about the sensitivity to each factor with that in mind.

We thank the reviewer for the time and efforts spent in reviewing this manuscript. Thanks for the recognition of the value of this paper on the coupled ice-ocean modelling community. The suggestion about splitting the influence of each factor into numerical and physical factors is good but like the reviewer said it is a huge amount of work. We will consider it in our future study.

In general, the language throughout could do with native-speaker tweaking by someone familiar with the specialist vocabulary. There's nothing that inhibits comprehension of anything that has been done or explained, but the current text is peppered with minor grammatical issues that give it a slightly awkward feel.

Thanks for the comments. We will address all points raised below as part of our revisions. Note that all the line numbers, section numbers, and Figure index in blue refer to the modified manuscript.

**Specific comments**

line15: "response is insensitive". Although the basic presence of some oscillatory behaviour is a constant as all of these factors are varied, since this work shows the amplitude, phase and sub-cycle variability of the oscillation varying significantly across

the different configurations, I think this sentence could easily be taken the wrong way by a reader so should be made more nuanced.

Thanks for the suggestion. We modified this sentence into "Sensitivity tests demonstrate that the oscillation feature is always present regardless of the choice of coupling interval, vertical resolution in the ocean model, tracer properties of immediately ungrounded cells by the retreating ice sheet, or the dependency of friction velocities to the vertical resolution, although the amplitude, phase and sub-cycle variability of the oscillation varied significantly across the different configurations." Relevant statements have been modified in the Discussion and Conclusion sections.

line35: further developments to NEMO that allow coupling to an ice sheet model (and the rest of an ESM) could be cited via Smith et al, JAMES 2021. https://doi.org/10.1029/2021MS002520

Thanks. New citation added.

line99: the sentence feels lost here - should it belong to either the preceding or succeeding paragraphs that describe the model rather than the experimental configuration?

Thanks for pointing it out. This sentence talks about the interface pressure at the ice-ocean interface. We modified this sentence as a separate paragraph and added it after talking about the three-equation parameterisation (Line 145) "In addition to the thermodynamic exchange at the ice-ocean interface, the pressure at the ice-ocean interface as an ocean boundary condition in the ocean model is calculated using the ice draft and a constant reference ocean density (1028 kg m$^{-3}$)."

line108: "Results indicate..." it's not clear to me if you mean previously published work or something you've done (which may or may not be described in this paper).

This result comes from the work we've done here with evidence in the following sentence. To make it clearer, we modified this sentence into "Additionally, it indicates that the barotropic timestep sizes has a dominant effect on simulating the basal melting (Fig. A1)."

line119: it's not clear if this sentence indicated that simulations here were unstable and failed to complete, or if this is just general advice for other users?

Some of the experiments in this study were unstable using DT100N60 and in such cases we used a finer baroclinic timestep size DT50N30 to complete the 100 years' run. It can also be an advice for other users to deal with the instability of their models. To make it clearer, we modified this sentence into "For the unstable experiments in this study, a smaller baroclinic timestep (DT50N30) was used for resolving instabilities and producing a simulation that runs to completion."

line125: "The water speed" - this is just the speed as used in the three equation-melt parameterisation, not water velocities in the model in general.

Thanks for pointing it out. I modified it into "The water speed in the three-equation parameterisation"

figure2: the oscillations would be clearer if the vertical axes were scaled to show them better. Could the large, linear adjustment in the first 20 years be shown in a separate panel on different axes to allow this? I'm not sure what is meant by the last sentence of the caption - what is the "nonphysical noise" that must be removed? The Ocean3 experiment doesn't appear in the main table of experiments and is not explained until much later in the paper - I'm not sure it helps to have those panels appear here.

Thanks for the suggestion. We modified the figure to be the new Fig. 3 as shown below without the first 20 years and add an appendix figure (Fig. A2) to show the whole 100 yr results.

[Figure]

Figure 3

[Figure]

Figure A2

We removed the noise in the signal to get a better consistency between the basal melting rate and the moving mean changes in grounded area and the maximum of the barotropic stream function. We don't know whether those noisy points are physical or not, so we removed the word "non-physical" here.

We add the description of Ocean 3 in Sec. 2.2. The comparison between CTRL and Ocean3 is discussed in Sec. 4.1. We think it is OK to add Ocean3 here for following discussions and saving space rather than adding an extra figure.

line162: Perhaps I should read the cited Gladstone'21 for the more details, but I'm unclear on what's allowed and what isn't in the passive thin cells. This sentence implies to me that there's potential for an ocean cell to unground because of changes in water pressure conditions in the thin layer due to the ocean, not the ice sheet. They're not just wet cells as a technical convenience in the model then...and there might be potential for some time dependence in the precise position of the grounding in this model in /all/ the

ISOMIP+ experiments, not just the ones where a change in ice sheet geometry is prescribed?

The reviewer is correct to note that changes in water pressure conditions can cause transient grounding/ungrounding in the ocean model independently of the forcing received from the ice model. Yes, the dry cells can unground by water inflowing, not only by the dynamical ice thinning. Water is allowed to flow into passive ("dry") cells, but not out from these cells, at least not until they have become active (i.e., "wet"). In regions where the ice is close to floatation and the gradients in ice geometry are small, we do indeed see some short timescale transient fluctuations in the grounded mask. This does not occur where ice geometry gradients are steeper, i.e., along the main central section of the grounding line. The reviewer is also correct to point out that a closer read of the Gladstone FISOC GMD paper (Gladstone et al., 2021) will clarify the wet/dry behaviour in ROMS. We have not changed the main text in response to this comment as the transient grounding/ungrounding does not affect the main features described in the current document, the reviewer hasn't specifically requested any changes, and readers may, if they wish, read more about this in the cited paper.

line166: "following MISOMIP1": I think it would be helpful to outline the protocol a little more for those unfamiliar with it. Perhaps the paragraph at line 95 belongs better here, and with a little more detail? I assume that aside from Ocean3 (whose protocol I don't think is explained anywhere) you're following the first part of the IceOcean1 experiment as the ice is forced to retreat, but that should be said - along with more on what IceOcean1 did and did not involve (eg. surface forcing, ice calving) - explicitly.

Thanks for the suggestion. We don't think the paragraph at line 95 belongs better here because this is the Result section for presenting the results rather than model description. We add 'Sec. 2.2 Experiment design' to provide more details about MISOMIP1 and Ocean 3. About the surface forcing, we add one sentence in Sec. 2.2 "No external forcing is applied at the surface of the open ocean, which means there is no atmospheric or sea-ice fluxes. A "WARM" forcing, as the only forcing, is applied within a 10 km restoring region near the ocean's northern boundary" 。

See the new Sec. 2.2 below:

"Each coupled model experiment in this study was run for 100 years, following Experiment IceOcean1r of MISOMIP1 (Asay-Davis et al., 2016). Like in IceOcean1r, experiments in this study does not include a dynamic calving, in which ice thickness is allowed to be zero without calving. Various configuration in each experiment can be seen in Table 1 and corresponding sections in Sec. 3.

We build our coupled model following the ISOMIP+ projects for stand-alone ocean models with ice-shelf cavities and the MISMIP+ projects for ice sheet models. Result of ISOMIP+ Ocean3 from Asay-Davis et al. (2016) using the same ocean model will be used as a comparison to the control experiment in this study (CTRL in Table 1).

The ocean model in the coupled system is initialised with a steady-state ice geometry from the ice sheet model and a ``COLD'' initial condition following Asay-Davis et al. (2016). No external forcing is applied at the surface of the open ocean, which means there is no atmospheric or sea-ice fluxes. A ``WARM'' forcing, as the only forcing, is applied within a 10 km restoring region near the ocean's northern boundary (yellow area in Fig. 2a), which is consistent with the warm ice shelf cavities in Amundsen and Bellingshausen Seas. The warm water is expected to reach the ice-shelf cavity within the first two decades and induce strong basal melting and subsequent rapid GL retreat.

In Ocean3, the stand-alone ocean model uses the same steady-state ice topography with the initial state of the coupled system, and is run for 100 years with an annually prescribed evolving ice geometry. The ocean is initialized with the WARM profiles, forced with the WARM profile in the same restoring region with CTRL and strong melting is expected to begin immediately as the sub-shelf circulation spins up. More details about MISMIP+ and ISOMIP+ can be seen in Asay-Davis et al. (2016)."

line171: why (these) four points/focus on year 44? The oscillation at the blue dot does seem qualitatively different from that at the green. Some more information about why - or what the distance from grounding line, or the thickness of the water column at the points are (is the ice ungrounding just a little before regrounding, or a lot?) might be interesting.

We choose four points to represent the different locations, i.e., open water, near the ice front, near the grounding line, and on the grounding line. Year 44 is just a random choice here.

The green points experienced the initial spin-up stage and reached its melt high around year 18 and then started to decrease in melt due to far away from the grounding line (see Fig. A3c below). The blue points experienced much higher melting immediately when it became ungrounded. That's why they are qualitatively different in the basal melting.

We add further information about the water column thickness and the distance from the initial grounding line as Fig. A3c. We also add some relative text in Line 192 "The green and blue points experience rapid increase in melting in the first few years but are largely different in magnitude (Fig. A3b). The green point experiences the initial spin-up and subsequent slow increase in melting during the intrusion of the warm water from the northern boundary, while the blue point experiences much higher melting immediately since becoming ungrounded. It suggests that the high melt rates occur near the GL within a short distance less than 6 km".

[Figure]

Figure A3

line176: it would be helpful to have the compass directions rather than just x and y marked on figures 1 and 3 if you're going to use them in the description, especially since MISOMIP chose to use left-right x as north.

Thanks for the suggestion. We confirmed the north direction used in MISOMIP (Asay-Davis et al., 2016) is same with what we defined as north in this paper (see Line 72) To make it clearer in figures, we added the compass directions in Fig. 1.

[Figure]

figure 3,4. The top row of figure 3 could be zoomed closer in to the grounding line to show more of the relevant detail. The corresponding top row of figure 4 seems redundant?

Thanks for the suggestion. We also changed the scale of the colour bar. Please see the modified new Fig. 4 (the old Fig. 3).

[Figure]

Figure 4

The significant difference in the basal melting mainly occurred at one or two rows of cells near the grounding line. That's why all the panels in the top row of the old Fig. 4 looked similar. We modified the color scale and make it more informative as new Fig. 5 shown below.

[Figure]

Figure 5

line179: I wasn't initially sure exactly what was meant by "discrete" here. I can see it's explained later, but saying "The discretisation of GL retreat" at the start might be less confusing.

Thanks for the suggestion. We modified it into "the discretisation of GL retreat"

line195: can you distinguish between the dependence on the gyre for its introduction of warmer water to the interface (eg a purely thermal effect) and its influence on the friction velocity used in the 3 equation melt parameterisation?

No, we can't distinguish them without some tightly artificially controlled experiments, which are outside the scope of the present study. The system has some strong nonlinearities that would make a linear decomposition to identify the cause and effect nonsensical. The gyre would act to both bring in more warm water and increase the friction velocity - driving more melt - which would increase the gyre. Other studies exist for exploring the decomposition, for example, Jourdain et al. (2019) where they explored the dynamic/thermodynamic decomposition that explains the differences in melt rates between the simulations with and without tides.

figure5: whilst all the intervals do show some oscillation, the variation in periodicity that results does seem like a really interesting result - a good example of why I think the paper would be more useful rephrased in terms of the variations that /have/ been found as the coupling mechanics vary, rather than the fact that some kind of oscillation is robust across all the techniques.

Thanks for the comment. See the modified text "The simulated mean melt rates (Fig. 6) indicate very little sensitivity to the coupling interval between 0.5 days and 3 months in the general trend, which is consistent with the similar sensitivity tests with coupling period ranging between 1 month and 1 year using NEMO-Elmer/Ice (Favier et al., 2019). However, experiment CDT90 does not show an obvious oscillation pattern compared with the other experiments, which implies that using a coarse coupling interval may lead to the loss of temporal detail in the coupled ice sheet/ocean response. It can also be seen in the tests with 6-month and 12-month coupling periods in Favier et al. (2019), in which the oscillation feature was obviously smoothed. Additionally, mild variations in periodicity and magnitudes are found as the coupling interval varies. Tests with coupling intervals of 5 days or less show more consistency, while tests with coupling intervals of 15, 30, 90 days show some difference in magnitudes and phases. CDT30 is closer than

CTRL (15 days) to the shorter coupling intervals, suggesting that there might be some cancelling effects in CDT30. Further study to understand the causes and nature of the impact of coupling intervals greater than 5 days would be of benefit to the coupled ice - ocean modelling community. Further studies to understand the causes and nature of the impact of coupling intervals greater than 5 days would be of benefit to the coupled ice - ocean modelling community."

line199: do the ice or ocean model timesteps also vary with coupling interval?

The ocean model timestep size did not change with different coupling intervals, while the ice model timestep size equals the coupling interval. We have explained it well in Sec. 2.5 "The coupling interval is set to be same with the timestep size in the ice component (15 days in the control experiment), while the time resolution of the ocean component is much finer (100 s). Semi-synchronous coupling is adopted here, in which the ice component has a larger time step than the ocean model. We set the coupling interval equal to the ice model timestep size (15 days in CTRL), while the baroclinic timestep in the ocean model is 100 s.".

line245: this is a nice result that I think might be worth making more of. Coarser resolution ocean models used for pan-Antarctic or global simulations that some climate modelling groups are starting to use may effectively be in the same regime as UstarIndep due to their viscosity and poor resolution of details of flow under the ice. These ROMS experiments are all at the MISOMIP 2km COM resolution - are there clearly weaker oscillations in any lower resolution TYP simulations submitted to MISOMIP?

There are two other groups (NEMO-ElmerIce and NEMO-UKESE1is) who submitted the TYP simulations. See the plot below. We added this figure as Fig. A7 and related text in Sec. 4.1 " Fig. A7 shows the simulated mean basal melting from contributing coupled models using both COM configuration and typical parameters (TYP configuration) with different horizontal resolution in ocean models. There are no obvious oscillation features in both COM and TYP simulations using NEMO-ElmerIce but the noise has been smoothed in TYP with a finer ocean resolution (1 km). The basal melting pattern from NEMO-UKESE1is-COM is pretty similar with CTRL in this study while the TYP simulation (light blue line, ocean horizontal resolution of ~8 km) shows much worse oscillations. However it is hard to say that the coarser ocean horizontal resolution led to the worse oscillations since there are some other different configurations between their COM and TYP configurations."

[Figure]

line253: I'm not totally clear on exactly what is being done with the tracer values in these two different approaches as explained here. Melt from the shelves does imply additional mass being added to the ocean, so here are we only considering changes in ocean volume caused by dynamic changes to the ice shelf draft?

Yes, the impact of basal melting is imposed on the ocean model through salt/heat flux (Galton-Fenzi et al., 2012). When melting occurs and freshwater should be added, we actually remove salt. This has the effect of not adding more mass to the ocean. The impact of changing ice draft, which modifies the cavity volume, is handled separately. In the real world the ocean must respond to changing ice draft through a modified horizontal flux divergence, e.g. bringing in water from nearby to fill an expanding cavity. In the model, the local water column volume simply increases, and we must decide how to handle tracer properties locally in this evolving water column.

To make it clearer, we added one sentence here "Note that the handling of tracer properties through ice draft change is separate from the way in which basal melting is implemented, and the latter is imposed on the ocean model through salt/heat fluxes (Galton-Fenzi et al., 2012). In response to the ice draft change, we simply change the volume of the water column without adding any fluxes".

We also modified the description about this in Sec. 2.4 Line 159 "Changes in water column thickness due to ice shelf thinning would be maintained through increased horizontal convergence/divergence in the ocean circulation in response to mass/volume changes. ROMS effectively introduces a source/sink term imposed by adding or removing heat or salt at the ice/ocean boundary. For example, when the ice shelf melts, the model removes salt/heat rather than adding freshwater volume. The circulation change in this case isa result from density changes rather than volume changes. The approach using a source/sink term of heat/salt transfer imposes a choice upon the ocean model: either conserving the volume integrals of tracer values (temperature and salt) or preserving the absolute values, (e.g., heat or freshwater). Here we will explore the effect of both options on the ocean circulation in a coupled system in Sec. 3.5."

line266: this is another result I can see being really useful to others developing coupled ocean-ice models that it would be good to highlight more, rather than the simple persistence of the meltrate oscillation - although it would be even better if I understood exactly what was being done in the two different approaches that are being compared (see comment above).

We agree. We highlight its importance in the Discussion (Sec. 4.2) and Conclusion section "As the ice draft (and hence water column thickness) evolves, a choice must be made about tracer conservation in the ocean model. Since the model default option to conserve the tracer volume integral introduces spurious tracer drift under a changing ice geometry, we locally conserve absolute tracer values when adjusting the ice draft to obtain a physically plausible tracer evolution."

line281: there do appear seem to be significant differences in the mean melt rate, amplitude and phase of the oscillation though. My take-home from the figure is that the value chosen to initialise dry cells /does/ matter somewhat to the system in general, opposed to what is in the text. Figure 9, as with the other timeseries plots, would show the important details more clearly I think if the large linear adjustment in the first 20 years were cut out and the y-axis scaled to show the detail of the more steady melt rate.

Thanks for the comment. We modified the figure to include the period from year 20 to year 100 as below:

[Figure]

We think it is good to see the spin-up stage, so we put the original plot in the appendix (Fig. A7).

We agree that these tests show differences in mean melt rate, amplitudes, and frequencies. To make it clearer, we modified the text as "A similar oscillation pattern can be seen in all these experiments with different amplitudes and frequencies (Fig. 10 showing the period between 20-100 years and Fig. A7 with the initial spin-up stage included), which suggests low sensitivity of melt oscillations to the initialisation of dry cell tracer values. However, initialisation of dry cell tracer values does have a non-trivial impact on melt rates. In general trend, WETDRY1 with cold and fresh water provides the highest melt while CTRL shows lowest melt rate. WETDRY3 shows similar amplitudes and periods in oscillation with CTRL while WETDRY2 with warmer water presents smallest amplitudes except for the last two decades. It indicates the oscillation feature can not be removed by using different initialisations of dry cells but the initialisation of dry cells would impact the oscillation pattern with different degrees."

In the discussion, we added "In the current study, we did not find a strong sensitivity of melt oscillations to our choice of how to handle dry cell tracer properties. This suggests the values assigned to the dry cells are not a strong influence on the melt oscillations, but it does affect the melt rates in general trend, magnitudes, and frequencies. Colder and fresher dry cells in WETDRY1 drive more overturning through buoyancy when becoming wet and lead to slightly higher basal melting than other experiments."

figure 10: should this come much earlier? We're first told about this result and the multi-model nature of the oscillation at line 46.

Agree. We moved this figure to the Introduction section as the new Fig.1.

line311: I don't understand what is meant by this sentence.

Thanks. We modified this whole paragraph into "It is still difficult to discriminate whether the oscillation feature is a fundamental emergent intrinsic physical feature of the coupled system, or a numerical artefact that arises through the combination of multiple processes."

line339: You've given some plausible reasons for why the moving geometry might amplify the melt rates and thus the amplitude of an oscillation, but it's not obvious why your amplifying factors don't simply result in a positive feedback and general increase in

melt and retreat rate, rather than leading to the cyclic strengthening and weakening observed. Could you talk through a hypothesised cycle?

Thanks for the comments. We add one paragraph before this line to describe our understanding of the hypothesised cycle.

"Here is our understanding of the hypothesised mechanism behind each cycle. The mean flow develops in response to the ice shelf melting (see Fig. 5e) whereby a single clockwise gyre under the entirety of the ice shelf forms in response to the enhanced melt-buoyancy-driven circulation feedback. The melt rate increases in response to the accelerating flow and the transport of warmer waters to the deeper part of the ice shelf. We suggest that at the point of maxim melting the development of a baroclinic instability in the flow draws energy from the mean flow potential energy, leading to the collapse of the main gyre and the formation of a much smaller circulation feature that is contained closely to the deep grounding zone (see Fig. 5f). The destruction of the main gyre during this phase leads to a substantial reduction in the melt rates due to a cooler and slower ocean, and also the lack of intrusion of relatively warm waters that can locally cool the ocean. However, the minimal amount of melting is sufficient to again start to drive a buoyancy driven flow that over time reestablishes the single gyre circulation feature and the process repeats. Our understanding of what leads to the development of the baroclinic instability is unclear but several studies have recently shown that in the presence of both steep topography and strong horizontal density gradients the production of an instability in the Antarctic Slope Front can lead to the transfer of energy from a strongly established mean flow to an eddying field resulting in substantial impacts on heat and freshwater transport in the ocean (e.g., Charlotte Huneke et al. (2019); Stern et al. (2015); Stewart and Thompson (2016)). The generation of these instabilities at least with the Antarctic Slope Front and the associated cross-shelf exchange are sensitive to model resolution and to the choice of eddy mixing parameters (Hattermann et al., 2014), and requires further investigation."

line362: The UKESM-ice paper (Smith et al, JAMES 2021) referred to above discusses this partitioning and shows that the non-conservation implications of their approach (which I think is similar to "preserving absolute tracer values" here, although I may have misunderstood) in their more realistic domain is considerable.

We think the reviewer misunderstood these two studies here. The UKESM-ice paper discussed the impact of non-conservation in ocean volume and heat caused by ice shelf coupling changes on the global mean sea level rise. In this study, we focused on non-conservation of tracer properties due to the stretching of the vertical coordinates as a response to the ice draft changes.

line399: as a conclusion, I think that simply stating that "this oscillation pattern is not sensitive to ..." is a bit misleading and underplays a set of results that are potentially very interesting to others developing their coupling schemes in this field.

Thanks for the comment. We rephased this sentence into "A series of sensitivity tests showed that the existence of this oscillation pattern was insensitive to the choice of coupling interval, vertical resolution of the ocean model, the initialisation of tracer properties of the dry cells, or the dependency of friction velocities to the vertical resolution." We put the rest of the conclusions into a separate paragraph and start with "While the existence of the melt oscillations is robust to our various model configurations, we find that our model choices have a non-trivial impact on mean melt and ocean circulation strength, which might be interesting to the coupled ice ocean system community.

line409: the last conclusion refers to a rather technical point that has only been introduced very late in the paper via one paragraph in the discussion, with no supporting results shown. Is it really the best point to end on - or needed here at all?

Thanks for the comment. We agree to delete the last sentence.

figure A2: as stated earlier, I think the difference in the form of oscillation at the green and blue dots is interesting and would be nice to see explored a little.

Please see my response to line171 above.

figure A5: the caption and legend labels don't really explain the difference between the panels and lines in this figure.

Thanks for pointing it out. See the modified Fig. A5 and caption "Stratification structure of experiment (a) N21 (b) N21E and (c) N11 with different vertical resolution. N is the vertical layers used in the ocean model."

[Figure]

line421: it might be nice to Acknowledge the general MISOMIP effort/community here, since this paper does require the existence of that protocol and the significance of the oscillation does partly rely on the fact that other contributors found it too - and you have a plot with their results (fig10)!

Thanks for your suggestion. We really appreciate the efforts from the MISOMIP community. We add one sentence here "We thank the MISOMIP community by sharing their basal melt rates using different coupled models, that provoked this study".

**References**

Asay-Davis, X. S., Cornford, S. L., Durand, G., Galton-Fenzi, B. K., Gladstone, R. M., Gudmundsson, G. H., Hattermann, T., Holland, D. M., Holland, D., Holland, P. R., Martin, D. F., Mathiot, P., Pattyn, F., and Seroussi, H.: Experimental design for three interrelated marine ice sheet and ocean model intercomparison projects: MISMIP v. 3 (MISMIP +), ISOMIP v. 2 (ISOMIP +) and MISOMIP v. 1 (MISOMIP1), Geosci. Model Dev., 9, 2471-2497, 2016.

Charlotte Huneke, W. G., Klocker, A., and Galton-Fenzi, B. K.: Deep Bottom Mixed Layer Drives Intrinsic Variability of the Antarctic Slope Front, Journal of Physical Oceanography, 49, 3163-3177, 2019.

Favier, L., Jourdain, N. C., Jenkins, A., Merino, N., Durand, G., Gagliardini, O., Gillet-Chaulet, F., and Mathiot, P.: Assessment of sub-shelf melting parameterisations using the ocean–ice-sheet coupled model NEMO(v3.6)–Elmer/Ice(v8.3), Geosci. Model Dev., 12, 2255-2283, 2019.

Galton-Fenzi, B. K., Hunter, J. R., Coleman, R., Marsland, S. J., and Warner, R. C.: Modeling the basal melting and marine ice accretion of the Amery Ice Shelf, Journal of Geophysical Research: Oceans, 117, 2012.

Gladstone, R., Galton-Fenzi, B., Gwyther, D., Zhou, Q., Hattermann, T., Zhao, C., Jong, L., Xia, Y., Guo, X., Petrakopoulos, K., Zwinger, T., Shapero, D., and Moore, J.: The Framework For Ice Sheet–Ocean Coupling (FISOC) V1.1, Geosci. Model Dev., 14, 889-905, 2021.

Hattermann, T., Smedsrud, L. H., Nøst, O. A., Lilly, J. M., and Galton-Fenzi, B. K.: Eddy-resolving simulations of the Fimbul Ice Shelf cavity circulation: Basal melting and exchange with open ocean, Ocean Modelling, 82, 28-44, 2014.

Jourdain, N. C., Molines, J.-M., Le Sommer, J., Mathiot, P., Chanut, J., de Lavergne, C., and Madec, G.: Simulating or prescribing the influence of tides on the Amundsen Sea ice shelves, Ocean Modelling, 133, 44-55, 2019.

Stern, A., Nadeau, L.-P., and Holland, D.: Instability and Mixing of Zonal Jets along an Idealized Continental Shelf Break, Journal of Physical Oceanography, 45, 2315-2338, 2015.

Stewart, A. L. and Thompson, A. F.: Eddy Generation and Jet Formation via Dense Water Outflows across the Antarctic Continental Slope, Journal of Physical Oceanography, 46, 3729-3750, 2016.

---

## Author Comment (AC4)

Responses to Reviewer #1

Review of "Evaluation of an emergent feature of sub-shelf melt oscillations from an idealised coupled ice-sheet/ocean model using FISOC(v1.1)-ROMSIceShelf(v1.0)-Elmer/Ice(v9.0)" by Chen Zhao, Rupert Gladstone, Ben Galton-Fenzi, David Gwyther and Tore Hattermann.

Recommendation: minor revision

Sub-shelf melt oscillations emerge from coupled ocean–ice-sheet simulations of the Marine Ice Sheet Intercomparision Project (MISOMIP), and this paper investigates the causes of these oscillations. This is a useful study for the ocean–ice-sheet modelling community. The paper is well written and the sensitivity tests make sense and are clearly analysed. I only have minor comments and I suggest to accept the manuscript once they have been considered.

We thank the reviewer Dr. Nicolas Jourdain for the time and efforts spent in reviewing this piece of work. The detailed comments are very much appreciated and will be of great help to improve the quality of this study. We will address all points raised below as part of our revisions. Note that all the line numbers and section numbers in blue refer to the modified manuscript.

Specific comments:

Abstract: it should be reminded that there is no external (atmosphere or sea ice) forcing in the MISOMIP experiments. This would help understand that it is somewhat surprising that an ocean oscillation emerges.

Thanks for the suggestion. But we don't think it is necessary. We already describe the oscillation as an emergent feature, which already implies that it is not forced externally. We make it clearer in the Sec. 2.2 (Line 86). "No external forcing is applied at the surface of the open ocean, which means there is no atmospheric or sea-ice fluxes. A "WARM" forcing, as the only forcing, is applied within a 10 km restoring region near the ocean's northern boundary". We choose to leave discussion about external forcing to the main paper and not the abstract.

L. 53, 152 and at other places: I am not a native speaker, but "couple" should probably be "coupled" (or "coupling" for some occurrences).

Thanks for pointing it out. We have corrected all the words "couple" into "coupled" or "coupling".

L. 97: Weddell and Ross cavities are usually classified as cold, not warm.

Thanks for pointing it out. We corrected it into "…with the warm ice shelf cavities in Amundsen and Bellingshausen Seas".

L. 128: "This parameterization" is a bit unclear.

We mean the three-equation parameterisation here and modified it into "The three-equation parameterisation is typically applied between the top model layer and the ice…"

L. 145-146 (and caption of Tab. 1): It is not clear to me what is the difference between

"conserving the volume integrals of tracer values (temperature and salt)" and "preserve the absolute values, (e.g. heat or freshwater)" as, for example, the volume integral of temperature directly gives the heat content when multiplied by ρcp. Furthermore, how exactly is imposed the conservation: additional flux at the surface? uniform T,S correction? Without this information, it is difficult to understand section 3.5.

Thanks for the comment. We don't impose a physical flux. For the basal melting we impose salt/heat fluxes on the ocean model (Galton-Fenzi et al., 2012). For the ice draft change we simply change the volume of the water column without adding any fluxes as such. When melting occurs and freshwater should be added, we remove salt. That's why we have this decision about how to handle tracer properties.

To clarify this, we add a couple of lines here "Changes in water column thickness due to ice shelf thinning would be maintained through increased horizontal convergence/divergence in the ocean circulation in response to mass/volume changes. ROMS effectively introduces a source/sink term imposed by adding or removing heat or salt at the ice/ocean boundary. For example, when the ice shelf melts, the model removes salt/heat rather than adding freshwater volume. The circulation change in this case is a result from density changes rather than volume changes. The approach using a source/sink term of heat/salt transfer imposes a choice upon the ocean model: either conserving the volume integrals of tracer values (temperature and salt) or preserving the absolute values, (e.g., heat or freshwater). Here we will explore the effect of both options on the ocean circulation in a coupled system in Sec. 3.5".

We also added a sentence at the end of Sec. 3.5 "Note that the handling of tracer properties through ice draft change is separate from the way in which basal melting is implemented, and the latter is imposed on the ocean model through salt/heat fluxes (Galton-Fenzi et al., 2012). In response to the ice draft change, we simply change the volume of the water column without adding any fluxes."

Please provide more details on the CTRL and Ocean3 experiments somewhere in section 2 or 3.1 (initial state, temperature and salinity restoring near the northern boundary, coupled models or ocean model with ice draft evolution, etc).

Thanks for the comment. We added one section "Sec. 2.2 Experiment design" for more details about MISOMIP1 and Ocean3.

"Each coupled model experiment in this study was run for 100 years, following Experiment IceOcean1r of MISOMIP1 (Asay-Davis et al., 2016). Like in IceOcean1r, experiments in this study does not include a dynamic calving, in which ice thickness is allowed to be zero without calving. Various configuration in each experiment can be seen in Table 1 and corresponding sections in Sec. 3.

We build our coupled model following the ISOMIP+ projects for stand-alone ocean models with ice-shelf cavities and the MISMIP+ projects for ice sheet models. Result of ISOMIP+ Ocean3 from Asay-Davis et al. (2016) using the same ocean model will be used as a comparison to the control experiment in this study (CTRL in Table 1).

The ocean model in the coupled system is initialised with a steady-state ice geometry from the ice sheet model and a ``COLD'' initial condition following Asay-Davis et al. (2016). No external forcing is applied at the surface of the open ocean, which means there is no atmospheric or sea-ice fluxes. A ``WARM'' forcing, as the only forcing, is applied within a 10 km restoring region near the ocean's northern boundary (yellow area in Fig. 2a), which is consistent with the warm ice shelf cavities in Amundsen and Bellingshausen Seas. The warm water is expected to reach the ice-shelf cavity within the first two decades and induce strong basal melting and

subsequent rapid GL retreat.

In Ocean3, the stand-alone ocean model uses the same steady-state ice topography with the initial state of the coupled system, and is run for 100 years with an annually prescribed evolving ice geometry. The ocean is initialized with the WARM profiles, forced with the WARM profile in the same restoring region with CTRL and strong melting is expected to begin immediately as the sub-shelf circulation spins up. More details about MISMIP+ and ISOMIP+ can be seen in Asay-Davis et al. (2016)."

Fig. 2: is the maximum of the barotropic stream function calculated under the ice shelf or all over the MISOMIP domain?

We calculated the maximum of the barotropic stream function under the whole MISOMIP domain. To make it clearer, we modified the related text into "The highest correlation coefficient between the basal melting and the maximum of the barotropic stream function under the whole domain (Fig. 3b) is 0.99 without a lag within both the 30 days and 1 day outputs". Note the previous Fig. 2 is now Fig. 3 in the revised draft.

Fig. 4: it would be easier to see the signal if the plots were showing anomalies with respect to the mean between year 63 and year 70.

Thanks for the suggestion. But we don't think it is necessary. The significant difference in basal melting across one cycle only occurred in one or two rows of cells where the GL retreated. That's why it looked nearly the same for the basal melting. We modified the color scale to make it look better. See the new figure below.

[Figure]

Figure 5

L. 185-186: what gyre are the authors referring to? Are these the gyres near the northern boundary or the gyre circulation within the ice shelf cavity?

We mean the gyre circulation within the ice shelf cavity. To make it clearer, we modified it into "2) the gyre circulation within the ice shelf cavity calculated as the strength of the barotropic streamfunction."

L. 193-195: I do not understand what the authors want to show with the barotropic circulation: any melt variation is associated with a change in barotropic circulation due to the modified horizontal density gradient and its role in the geostrophic balance (see Jourdain et al., JGR, 2017).

Thanks for the comments. We want to say that the basal melting is very much correlated with the process that we already highly suspected is driving the melt. We modified this sentence into "There is a high correlation (0.99) with no lag between the gyre circulation and basal melting (see Fig. 3b)".

L. 199-204: it is not so much the melt rate that is insensitive to the coupling period (it is actually smoothed for 6-mont h and 12-month coupling periods in Favier et al. 2019), it is the ice-sheet dynamics. Fig. 5 should therefore include another panel to show the ice sheet response (e.g. volume above floatation).

The only way the ocean impacts on the ice dynamics is through basal melting. So if melting is consistent across runs it is reasonable to assume ice dynamic behavior will be too. In this sentence, we made a statement about the sensitivity of general trend in basal melting to the choice of coupling interval rather than talking about the oscillation features. After this statement, we mentioned that CDT90 shows a smoothed oscillation pattern. To make it clearer, we added another sentence at the end "The simulated mean melt rates (Fig. 6a) and the ice volume above floatation (Fig. 6ab) indicate very little sensitivity to the coupling interval between 0.5 days and 3 months in the general trend. This is consistent with sensitivity tests with coupling periods ranging between 1 month and 1 year using NEMO-Elmer/Ice (Favier et al., 2019), in which the mean cavity melt rate seen by Elmer/Ice shows very little sensitivity to the coupling period. However, experiment CDT90 does not show an obvious oscillation pattern compared with the other experiments, which implies that using a coarse coupling interval may lead to the loss of temporal detail in the coupled ice sheet/ocean response. It can also be seen in the tests with 6-month and 12-month coupling periods in Favier et al. (2019), in which the oscillation feature was obviously smoothed. Additionally, mild variations in periodicity and magnitudes are found as the coupling interval varies. Tests with coupling interval of 5 days or less show more consistency, while tests with coupling intervals of 15, 30, 90 days show differences in magnitudes and phases. CDT30 is closer than CTRL (15 days) to the shorter coupling intervals, suggesting that there might be some cancelling effects in CDT30. Further study to understand the causes and nature of the impact of coupling intervals greater than 5 days would be of benefit to the coupled ice - ocean modelling community."

We added another panel to show the ice sheet response and see new Fig. 6 below.

[Figure]

Figure 6 (a) Simulated mean melt rates and (b) ice volume above floatation with different coupling interval. The inset box in (b) is the zoomed in period between year 60 to year 70.

L. 201-202: While I appreciate that FISOC is flexible, this sentence comes out of the blue and I would remove it.

Removed.

Fig. 6: The vertical resolution seems to have an effect on the melt oscillation period (e.g. compare orange to black curves).

Yes, similar with other tests with different coupling interval, different initialisation of tracer properties of the dry cells, or the dependency of friction velocities to the vertical resolution, they all affect the amplitude and period of the melt oscillation at different degrees. We have mentioned it in Sec. 3.3 (Line 246) "A similar oscillation pattern existed in all of the experiments related with vertical resolution, but showed different frequencies and amplitudes. The outcomes of these experiments demonstrate that emergence of the basal melt oscillation does not depend on the vertical resolution of the ocean model.".

L. 238: fu* should be u*

Modified.

L. 239: if melt is independent from u*, what equivalent constant u* value is applied?

In the 'three-equation parameterization' equation, the exchange velocity can be either assumed constant or assigned a functional dependence on the friction velocity u*. In UstarIndep, we adopted a constant $\Upsilon_T$ (thermal exchange velocity at the ice-ocean interface) and $\Upsilon_S$ (salinity exchange velocity at the ice-ocean interface) to remove the dependence of exchange velocity to u*. To make it clearer, I added the following sentences in Line 263:

"2) UstarIndep, in which we used constant values of thermal and salinity exchange velocities at the ice-ocean interface (T = $1 \times 10^{-4}$ m s$^{-1}$, S = $5.05 \times 10^{-7}$ m s$^{-1}$). The chosen values match those used by Hellmer and Olbers (1989), and are approximately equivalent to a constant friction velocity of ~0.01 m s$^{-1}$".

Section 4.1: more information is needed: do all these models have the same ocean and/or ice-sheet resolution?

We added one sentence in Line 328 to make it clear. "All the contributing ocean models used the same horizontal resolution of 2 km while the ice modes used different horizontal resolution near the grounding line ranging from 200 m to 1 km."

Fig. 11: anomalies with respect to the entire period would be better.

Thanks for the suggestion. See modified figure below.

[Figure]

Figure 12 XZ sections of anomalies of overturning streamfunction near the grounding line from CTRL (top row) and Ocean3 (bottom row) around one oscillation cycle. Anomalies are calculated with respect to the whole cycle. The chosen time points are

shown with red points in Fig. 3.

L. 307 and 349-342 and 401: I do not see why the grid direction would matter, the issue of having discrete grounding line retreat will remain whatever the grid direction. I don't pretend that it won't make any difference, but I do not see why it would make oscillations disappear (for example, the ice slopes will still be affected by the grounding line motions). Instead of rotating the grid, I would suggest increasing the ocean resolution.

Thanks for the comment. We don't agree that the rotation of the grid would not remove the oscillations. The oscillations feature a correlation between the ungrounding of a row of grid cells and enhanced melting and circulation strength. The orientation of the grid and the design of the experiment (such that the central part of the GL is aligned with the grid) allow this ungrounding of a whole row of grid cells to occur approximately together. If the grid were rotated, the experiment design would not encourage the ungrounding of a whole row of cells. Instead, it could be that cells unground one at a time. We do not know whether the melt oscillations would then occur the same as in the current set up, with reduced strength, or not at all. Reduced strength seems most likely since smaller scale discrete ungrounding would still occur. A grid rotated to about 45 degrees would potentially allow a different pattern of ungrounding to appear. We acknowledge that an increased ocean model resolution may reduce this effect, but only if it can resolve a more complex grounding line geometry which is no longer aligned with the model grid.

To make it clearer, we modified those texts as below:

L307 "The fact that they occur only in simulations in which the GL moves, together with the close relation between GL retreat and mean melt, strongly suggests that the melt oscillations are driven by the discretized ungrounding that occurs on a structured grid that is aligned with the GL. The grid orientation and the experiment design in this study guarantee the central part of the GL aligned with the grid, which allows the ungrounding of a whole row of grid cells to occur approximately together. A grid rotated to about 45 degrees would potentially allow a different pattern of ungrounding to appear. If the grid were rotated to about 45 degrees, the experiment design would not encourage the ungrounding of a whole row of cells and cells may unground one at a time instead. We do not know whether the melt oscillations would then occur the same as in the current set up, with reduced strength, or not at all. Reduced strength seems most likely since smaller scale discrete ungrounding would still occur. A further test with a rotated grid in the ocean model might help to diagnose the potential numerical issues associated with coupled grounding line retreat processes."

L339-342 "Our results however also suggest that the pattern of ungrounding is controlled by the discretisation of the coupled system (primarily the ocean grid) and future work should investigate the use of a grid rotated to about 45 degrees to test the sensitivity. In a real-world simulation, in which the GL is not aligned with the model grid, do these melt oscillations still occur in the similar way? We also recommend future studies by employing finer resolution near the GL in the ocean model and quantifying the impacts of finer resolution and grid rotation to determine whether the time-mean melt in the current study is affected by numerical artefact."

L401: We think it is fine to say "Future studies with a higher horizontal resolution and a rotated ocean model grid will help further quantify the impact on this oscillation feature, and determine whether the melt oscillation is a numerical model artefact."

L. 323: buoyant plume speed…. and speed associated with the horizontal density gradient.

Thanks for the suggestion. We modified "buoyant plume speed" into "speed associated

with the horizontal density gradient".

L. 399: "not sensitive" -> "not very sensitive".

Modified the sentence into "the existence of this oscillation pattern was insensitive to the choice of …"

**References**

Asay-Davis, X. S., Cornford, S. L., Durand, G., Galton-Fenzi, B. K., Gladstone, R. M., Gudmundsson, G. H., Hattermann, T., Holland, D. M., Holland, D., Holland, P. R., Martin, D. F., Mathiot, P., Pattyn, F., and Seroussi, H.: Experimental design for three interrelated marine ice sheet and ocean model intercomparison projects: MISMIP v. 3 (MISMIP +), ISOMIP v. 2 (ISOMIP +) and MISOMIP v. 1 (MISOMIP1), Geosci. Model Dev., 9, 2471-2497, 2016.

Galton-Fenzi, B. K., Hunter, J. R., Coleman, R., Marsland, S. J., and Warner, R. C.: Modeling the basal melting and marine ice accretion of the Amery Ice Shelf, Journal of Geophysical Research: Oceans, 117, 2012.

Hellmer, H. H. and Olbers, D. J.: A two-dimensional model for the thermohaline circulation under an ice shelf, Antarctic Science, 1, 325-336, 1989.

---

## Author Response (AR2)

**Responses to the Editor**

Many thanks for the Editor's suggestions on our manuscript. All minor remarks are accepted and modified in the revised manuscript.

-L82 : « build » should use the past tense : « built »

Done

-L123 : « it indicates » should be « they indicate » as it refers to the results, right ?

Yes, it refers to the results. Done.

-L321 : « for benefit of application on a real world » : I do not understand what this means, please rephrase ; do you simply mean «for real-world applications » ?

Done.

-L328 : « All the contributing models» should be « All contributing ocean models »

Done.

-L329 : « ice modes » should be « ice models », I think

Done.

-L394 : I suggest you remove « Our understanding of »

Done.